# LeDex: Training LLMs to Better Self-Debug and Explain Code

**Nan Jiang**[1]* **Xiaopeng Li**[2] **Shiqi Wang**[2] **Qiang Zhou**[2] **Soneya Binta Hossain**[3]*
**Baishakhi Ray**[2] **Varun Kumar**[2] **Xiaofei Ma**[2] **Anoop Deoras**[2]
[1]Purdue University [2]AWS AI Labs [3]University of Virginia
jiang719@purdue.edu
{xiaopel,wshiqi,zhouqia,rabaisha,kuvrun,xiaofeim,adeoras}@amazon.com
sh7hv@virginia.edu

## Abstract

In the domain of code generation, self-debugging is crucial. It allows LLMs to refine their generated code based on execution feedback. This is particularly important because generating correct solutions in one attempt proves challenging for complex tasks. Prior works on self-debugging mostly focus on prompting methods by providing LLMs with few-shot examples, which work poorly on small open-sourced LLMs. In this work, we propose **LeDex**, a training framework that significantly improves the self-debugging capability of LLMs. Intuitively, we observe that a chain of explanations on the wrong code followed by code refinement helps LLMs better analyze the wrong code and do refinement. We thus propose an automated pipeline to collect a high-quality dataset for code explanation and refinement by generating a number of explanations and refinement trajectories from the LLM itself or a larger teacher model and filtering via execution verification. We perform supervised fine-tuning (SFT) and further reinforcement learning (RL) on both success and failure trajectories with a novel reward design considering code explanation and refinement quality. SFT improves the pass@1 by up to 15.92% and pass@10 by 9.30% over four benchmarks. RL training brings additional up to 3.54% improvement on pass@1 and 2.55% improvement on pass@10. The trained LLMs show iterative refinement ability and can keep refining code continuously. Lastly, our human evaluation shows that the LLMs trained with our framework generate more useful code explanations and help developers better understand bugs in source code.

## 1 Introduction

Code generation has become a crucial research task to automatically generate source code based on natural language description [1–4]. Although the recent Large Language Models (LLMs) have shown impressive capability in code generation, generating the correct code for a complex problem in single attempt is still challenging [5–12]. This is expected because even for human developers, completing a hard programming problem might need multiple rounds of trial-and-error debugging. Self-debugging capability that allows LLMs to retrospect the incorrect code and make changes to resolve the errors is becoming increasingly important besides their code generation ability.

Existing works [13, 14] investigate off-the-shelf LLMs in the scale of Codex (`code-davinci-002`) [1], GPT-3.5 and GPT-4, and show that these LLMs can self-debug the wrong code they generated via prompting methods in a pipeline of code generation and self-refinement as shown in Figure 1. The

---

*Work done while interning at AWS AI Labs

38th Conference on Neural Information Processing Systems (NeurIPS 2024).

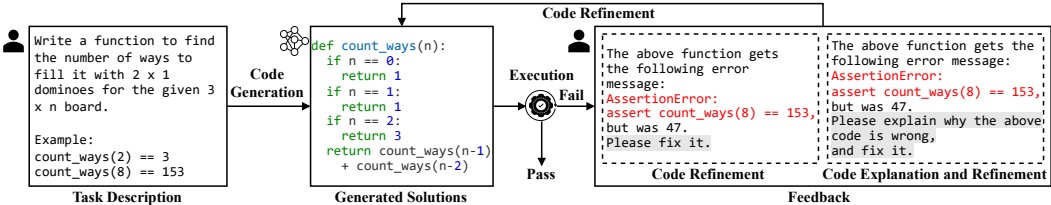

Figure 1: Pipeline of letting LLM generate code and self-debug.

user first queries the LLM for a solution for the given programming task and the initial solution from the LLM is verified by executing them against the given unit tests. If the solution passes all the unit tests, it is considered correct. Otherwise, the user collects the unit test feedback and forms a new query to ask the LLM for a refinement. Such a process can iterate until the LLM generates a correct solution or reaches the maximum number of iterations. There are different prompt designs when asking for refinement [13]. Compared with directly asking for a refined solution (referred to as "Code Refinement" in the feedback block), asking LLMs to provide an explanation of the wrong solution and then refine it in a chain-of-thought manner (referred to as "Code Explanation and Refinement" in the feedback block) helps it to better understand the unit test feedback and increases the success rate of providing refined solutions (details in Appendix A.1).

However, how to improve LLMs' self-debugging capability remains under-explored, especially given the fact that open-sourced LLMs such as StarCoder [11] and CodeLlama [12] have limited self-refinement performance. For example, the StarCoder-15B model is only able to refine 4.43% wrong solutions for problems from the MBPP benchmark [3], in contrast, GPT-3.5-Turbo can refine 28.90% under the same setting (details in Appendix A.1). Such limited self-refinement ability motivates the need to better train LLMs to take feedback to explain and self-refine the wrong code. Although important, an essential challenge of training LLMs to explain and refine wrong code is the lack of training data, especially high-quality code explanation data. Previous work has explored Imitation learning from Language Feedback (ILF) [15], which trains LLMs with human-annotated explanation, yet, such an approach is not scalable and the LLMs also do not obtain the ability to explain code.

In this work, we propose LEDEX, an automated pipeline to collect a high-quality dataset for code explanation and refinement by generating explanation and refinement trajectories, followed by filtering through execution verification. LEDEX then leaverages the collected data, using supervised fine-tuning (SFT) to significantly improve LLMs' ability to explain and refine incorrect code. Additionally, LEDEX applies reinforcement learning (RL) with a novel reward design that accounts for explanation semantics and unit test success, leading to better code explanations and corrections. In summary, this work contributes the following:

- We introduce LEDEX, a scalable framework comprising automated data collection, data validation, supervised fine-tuning, and reinforcement learning with novel reward mechanisms to enhance LLMs' self-debugging capabilities, resulting in more accurate code refinements and insightful code explanations.
- We experiment LEDEX on three backbones (StarCoder-15B, CodeLlama-7B, and CodeLlama-13B) using code refinements and explanations, initially collected from GPT-3.5-Turbo. Supervised fine-tuning notably boosts the models' ability to diagnose and correct faulty code, achieving up to a 15.92% improvement in pass@1 and a 9.30% increase in pass@10 across four benchmarks.
- LEDEX's reinforcement learning on top of SFT, uses a novel reward function that incorporates unit test outcomes and semantic analysis of incorrect code explanations. This further enhances performance, with improvements of up to 3.54% in pass@1 and 2.55% in pass@10.
- LEDEX is model-agnostic; notably, CodeLlama-7B trained on data gathered from CodeLlama-34B or even itself achieves up to 8.25% and 2.14% gains in pass@1 and pass@10, demonstrating the generalizability of the approach without reliance on GPT-3.5-Turbo.

## 2   Approach

Figure 2 shows the overview of LEDEX, including the collection of high-quality code explanation and refinement data, and the training methods. LEDEX first collects a code explanation and refinement

dataset by querying from pre-trained or instruct models and verifying its responses with execution feedback to filter and obtain high-quality explanation and refinement data (steps 1 and 2 in Figure 2, Section 2.1). Then the high-quality dataset is used for supervised fine-tuning (step 3 in Figure 2, Section 2.2), which significantly improves the model's performance in explaining the bug and refining the code. Reinforcement learning with execution feedback is used to further guide the model to generate higher quality responses and boost the model performance (step 4 in Figure 2, Section 2.3).

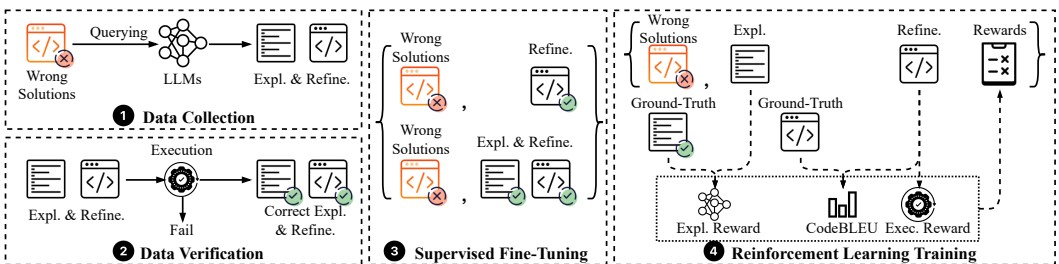

Figure 2: Overview of LEDEX.

## 2.1 Data collection and verification

We use MBPP [3] (only use the 374 problems in the training set during training), APPS [4] (only use the 5,000 problems in the training set) and CodeContests [2] as our base training datasets, which contain programming problems and solutions collected from various platforms. While they are helpful for training LLMs for code generation, they neither contain enough wrong solutions nor the explanation and refinement of them. To collect more wrong solutions, we prompt the pre-trained LLMs (i.e., StarCoder and CodeLlama) accordingly with 3-shot examples to sample 20 solutions (temperature set to 1.0) per problem from MBPP's training set, APPS's training set, and CodeContests. We then run these generations against test cases to select the wrong solutions that fail any test cases.

Table 1 shows the number of correct (passing all the unit tests) and wrong (failing any unit test) solutions sampled for each dataset. For each wrong solution, we need an explanation of the wrong code and a correct refinement to build the code explanation and refinement dataset. We prompt pre-trained or instruction-LLMs with the problem description, wrong solution, and execution feedback (either error message or failed test case) to ask for an explanation and refinement. We experimented with GPT-3.5-Turbo, CodeLlama-34B, and CodeLlama-7B for data collection. We take GPT-3.5-Turbo as the example in this section, and an example with it is shown in Appendix A.2. We study the generalization of this data collection with different LLMs in Section 4.3.

Table 1: Number of unique, correct, wrong solutions sampled from pre-trained LLMs, as well as the number of correct refinement generated by GPT-3.5-Turbo and its refinement rate on each dataset.

| Dataset | #Unique Solutions | #Correct Solutions | #Wrong Solutions | #Correct Refinement | #Refinement Rate |
|---|---|---|---|---|---|
| MBPP Training (374) | 9,500 | 4,706 | 4,794 | 2,203 | 45.95% |
| APPS Training (5,000) | 44,108 | 27,736 | 16,372 | 6,419 | 39.21% |
| CodeContest (6,627) | 51,134 | 31,520 | 19,614 | 5,113 | 26.07% |

As LLMs may provide wrong explanations or refinements, we cannot blindly take them as training data. Thus, we verify the refinements by running them against the test cases again, and only those passing all the test cases are considered correct refinements. For explanation, we consider the explanations along with the correct refinements as correct. Overall, for example with GPT-3.5-Turbo, we get 13,735 correct explanations and refinements: 2,203 for MBPP, 6,419 for APPS, and 5,113 for CodeContests. This verification step is crucial to guarantee the quality of the automatically collected code explanation and refinement dataset.

## 2.2 Supervised fine-tuning

We form the fine-tuning data in an instruction-following format similar to StarChat [16], where the user input is enclosed by <|user|> and <|end|>, while LLM's answer is enclosed by <|assistant|> and <|end|> in the chat history. Moreover, to alleviate the limited amount of data, we augment the

fine-tuning data by using two different instructions: providing the task description, the initial wrong code, and execution feedback, asking for (1) a refinement directly, or (2) an explanation of the wrong code and then a refinement in a chain-of-thought manner. Examples are given in Appendix A.2

During supervised fine-tuning, although we include the wrong solutions as LLM's initial answer in the chat history, we do not calculate the loss for this part since we do not want the LLM to intentionally generate those wrong solutions. They are just provided as context for code explanation and refinement if the LLM indeed makes mistakes in real use cases.

## 2.3 Reinforcement learning

Reinforcement learning is widely used to further improve the quality of LLM's generated outputs [17–20]. Through the RL framework, the LLM is optimized by using an algorithm to update the weights using both success and failure trajectories and maximize the rewards of its outputs. To train the fine-tuned LLMs to generate better code explanations and more correct code refinements, we design the rewards considering both parts.

### 2.3.1 Refinement score

To train LLM to refine code, the correctness of the refinement is the main goal, which can be measured by its code similarity to the ground truth, as well as the execution result. We use CodeBLEU score as metrics for code similarity and unit test passing rate as metrics for execution results.

Given a wrong solution $w$, the set of correct and wrong (failed) refinements are notated by $R_c^w$ and $R_w^w$. For any refinement $r$, we calculate its CodeBLEU score and the unit test passing rate as follows:

$$S_{cb}(r) = \frac{1}{|R_c^w|} \sum_{r_c \in R_c^w} \text{CodeBLEU}(r, r_c); \qquad S_{ut}(r) = \frac{|T_p(r)|}{|T|}$$

$S_{cb}$ is the average CodeBLEU score between a given refinement and all the correct refinements. $S_{ut}$ is the fraction of the number of passed unit test cases ($T_p$) when running the refined code $r$, over the total number of unit test cases ($T$) provided for this problem in the dataset.

In Figure 3, the x-axis is the scores of certain metrics, and the y-axis is the number of training data with a certain score (same for other figures in Figure 3). Thus, Figure 3 (a) shows the frequency distribution of each score of $S_{cb}$, with blue bars referring to training data with correct refinements, and orange bars referring to that with wrong refinements. The distribution of $S_{ut}$ is shown in Figure 3 (b) where the correct refinements definitely pass all the test cases and can be separated from the wrong ones.

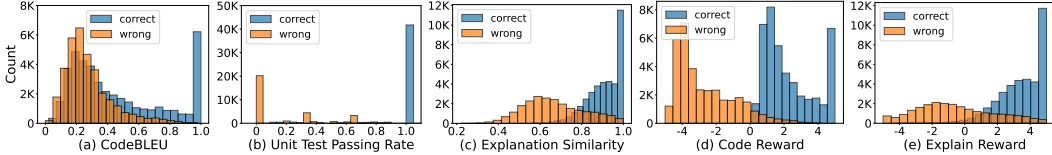

Figure 3: The CodeBLEU scores, unit test cases passing rate, sentiment similarity of wrong code explanations, final refinement code reward, and the explanation reward of the **training data**.

### 2.3.2 Explanation score

In our dataset, there are wrong code explanations along with code refinement, whose quality may not be perfectly reflected by the code quality. A correct code explanation may also be followed by incorrect refinement, thus, it is necessary to consider the explanation in the reward. The code explanations followed by correct refinements are treated as ground truth, notated by $E_c^w$. We calculate the average sentiment similarity [21, 22] between the explanation embedding $e$ and corresponding embeddings in ground truth as

$$S_{ex}(e) = \frac{1}{|E_c^w|} \sum_{e_c \in E_c^w} \text{CosSim}(\text{RoBERTa}(e), \text{RoBERTa}(e_c))$$

The distribution of $S_{ex}(e)$ is shown in Figure 3 (c).

### 2.3.3 Reward design

Given a pair of explanation and refinement, i.e., $(e, r)$, the reward of the generated code refinement and the code explanation is designed as:

$$\mathcal{R}(r) = 5 \cdot (S_{cb}(r) + S_{ut}(r)) - 5; \qquad \mathcal{R}(e) = \frac{50 \cdot S_{ex}(e) - 35}{3}$$

This code reward $\mathcal{R}(r)$ is the average of CodeBLEU score and the unit test passing rate, and since both $S_{cb}(r)$ and $S_{ut}(r)$ are scored in the range of [0, 1], this equation makes the reward of code refinement in the range of [-5, 5]. Figure 3 (d) shows the distribution of the code refinement reward on the training dataset, which mitigates the overlap issue between correct and wrong outputs with CodeBLEU score alone as illustrated in Figure 3 (a). It also makes the reward distribution continuous, addressing the discreteness problem of only using unit test passing rate.

For the design of explanation reward $\mathcal{R}(e)$, We observe from Figure 3 (c) that the explanation sentiment similarities of the training data mostly lie in the range of [0.4, 1.0], thus, we project the range of [0.4, 1.0] to [-5, 5] and treat 0.7 as the borderline (projected to 0 correspondingly) of good or bad explanations. Figure 3 (e) shows the distribution of the wrong code explanation reward on the training dataset. The distribution shows that there could be a good or correct code explanation followed by a wrong code refinement, where assigning a high or low reward to the entire output is not reasonable. This leads to our PPO [23] algorithm with code refinement and explanation rewards considered separately. Due to space limit, the PPO algorithm is shown in Appendix A.3.

## 3 Experimental setup

For supervised fine-tuning, we fine-tune three LLMs (StarCoder-15B, CodeLlama-7B, and CodeLlama-13B) using the correct initial solutions and correct refinements collected from the MBPP training set, APPS training set, and CodeContests. The model is fine-tuned for two epochs, using a batch size of 128. The optimizer is AdamW [24] with learning rate set to $2e^{-5}$. The learning rate is adjusted using a warmup of 500 steps and then decayed following a cosine scheduler.

We further train supervised fine-tuned LLMs with reinforcement learning using the PPO algorithm. The reinforcement learning training data is all the initial solutions and collected refinement on the MBPP and APPS training set. The learning rate is $2e^{-6}$, and the batch size is set to 64. We implement reinforcement learning training based on the TRL [25] library. Both the supervised fine-tuning and reinforcement learning are conducted on 8 NVIDIA A100 GPUs, each with 40GB of memory.

## 4 Result

To evaluate the effectiveness of LEDEX, we study the following research questions (RQs) regarding code generation and code refinement capability, iterative refinement ability, approach generalizability, and the quality of the generated code explanations.

### 4.1 RQ1: Code generation and refinement capability

We evaluate the models trained with LEDEX for their code explanation and refinement ability using four benchmarks: MBPP [3], HumanEval [1], MBPP$^+$ [26], and HumanEval$^+$ [26]. We use pass@k [1] and success refinement rate as the evaluation metric. For the generation of the initial solutions, the models sample 100 solutions per task in the benchmarks (temperature set to 0.8), which are run against the provided test cases. For every incorrect solution that fails any test case, we let the models sample one refinement (and one explanation).

### 4.1.1 Pass@k

Table 2 presents the pass@k results across four benchmarks. Overall, fine-tuning the LLMs with our curated dataset of code explanations and refinements leads to substantial improvements in both pass@1 and pass@10 for all three model architectures. For StarCoder-15B and CodeLlama-13B,

Table 2: Pass@k of initial and refined solutions on four benchmarks. Each backbone's best performance on every benchmark is bolded.

| Models | Approaches | | MBPP | | Humanval | | MBPP$^+$ | | HumanEval$^+$ | |
|---|---|---|---|---|---|---|---|---|---|---|
| | | | pass@1 | pass@10 | pass@1 | pass@10 | pass@1 | pass@10 | pass@1 | pass@10 |
| StarCoder-15B | Prompt. | Init. | 37.70 | 69.60 | 30.34 | 63.21 | 34.90 | 60.85 | 26.16 | 56.77 |
| | | Refine | 40.64 | 71.04 | 34.05 | 65.42 | 40.48 | 66.59 | 30.73 | 61.11 |
| | | Expl. + Refine | 40.37 | 71.60 | 33.96 | 67.20 | 39.23 | 64.67 | 30.09 | 60.72 |
| | LEDEX SFT | Init. | 47.41 | 70.27 | 35.05 | 66.93 | 43.28 | 62.48 | 30.01 | 59.77 |
| | | Refine | 56.66 | 75.78 | 47.32 | 77.06 | 53.53 | 71.55 | 43.16 | 71.74 |
| | | Expl. + Refine | 57.11 | 76.70 | 47.37 | 77.16 | 53.83 | 72.19 | 43.54 | 72.61 |
| | LEDEX RL | Init. | 48.62 | 70.68 | 39.45 | 70.16 | 44.94 | 63.92 | 35.05 | 64.41 |
| | | Refine | 58.00 | **78.12** | **52.25** | **80.80** | 54.13 | 71.71 | 46.11 | 74.07 |
| | | Expl. + Refine | **58.19** | 77.96 | 51.67 | 80.79 | **54.29** | **71.93** | **46.26** | **74.24** |
| CodeLlama-7B | Prompt. | Init. | 38.21 | 67.24 | 34.27 | 69.60 | 37.18 | 61.23 | 27.40 | 60.81 |
| | | Refine | 43.98 | 71.94 | 39.20 | 74.14 | 42.97 | 66.89 | 31.84 | 65.08 |
| | | Expl. + Refine | 43.42 | 72.09 | 40.13 | 74.95 | 42.46 | 67.41 | 32.49 | 66.58 |
| | LEDEX SFT | Init. | 48.87 | 70.89 | 36.99 | 69.95 | 42.97 | 62.69 | 30.76 | 62.52 |
| | | Refine | **58.07** | 77.34 | 52.65 | 80.71 | 51.64 | 71.04 | 46.61 | 74.43 |
| | | Expl. + Refine | 57.98 | 77.92 | 52.98 | 82.22 | 51.55 | 70.94 | 47.62 | 75.54 |
| | LEDEX RL | Init. | 46.54 | 71.54 | 39.38 | 71.84 | 41.46 | 63.68 | 33.95 | 65.98 |
| | | Refine | 57.35 | 78.12 | 54.41 | 82.55 | 52.11 | **71.88** | 48.73 | 77.17 |
| | | Expl. + Refine | 57.92 | **78.97** | **55.84** | **84.14** | **52.90** | 71.80 | **50.04** | **78.25** |
| CodeLlama-13B | Prompt. | Init. | 42.88 | 70.85 | 37.11 | 74.69 | 38.93 | 62.26 | 30.15 | 66.27 |
| | | Refine | 49.68 | 75.85 | 45.78 | 81.07 | 46.22 | 70.14 | 37.62 | 72.68 |
| | | Expl. + Refine | 49.97 | 76.39 | 45.90 | 81.18 | 45.77 | 70.48 | 38.36 | 73.84 |
| | LEDEX SFT | Init. | 52.43 | 73.66 | 41.65 | 73.61 | 43.67 | 62.63 | 35.29 | 68.49 |
| | | Refine | 61.78 | 79.96 | 58.41 | 83.47 | 55.58 | 73.41 | 51.35 | 77.84 |
| | | Expl. + Refine | 61.59 | 80.21 | 57.76 | 84.57 | 54.59 | 72.15 | 51.32 | 78.84 |
| | LEDEX RL | Init. | 51.19 | 73.16 | 45.45 | 74.79 | 45.49 | 62.81 | 39.27 | 69.29 |
| | | Refine | **61.98** | 79.95 | **61.71** | 84.58 | **57.89** | **73.48** | **56.68** | 80.89 |
| | | Expl. + Refine | 61.63 | **80.27** | 61.66 | **86.23** | 56.62 | 72.04 | 56.57 | **81.77** |

Table 3: Overall pass@k on MBPP & HumanEval and MBPP$^+$ & HumanEval$^+$. Blue or red numbers show the improvement or deterioration: SFT is compared to prompting, and RL is compared to SFT.

| StarCoder-15B | MBPP & HumanEval | | | | MBPP$^+$ & HumanEval$^+$ | | | |
|---|---|---|---|---|---|---|---|---|
| | LEDEX SFT | | LEDEX RL | | LEDEX SFT | | LEDEX RL | |
| | Refine | Expl. + Refine | Refine | Expl. + Refine | Refine | Expl. + Refine | Refine | Expl. + Refine |
| pass@1 | 54.35 +15.34 | 54.71 +15.92 | **56.78** +2.43 | 56.58 +1.87 | 49.31 +12.80 | 49.64 +14.13 | 50.87 +1.56 | **51.02** +1.38 |
| pass@10 | 76.23 +6.58 | 76.81 +6.30 | **78.78** +2.55 | 78.66 +1.58 | 71.63 +7.27 | 72.36 +9.30 | 72.67 +1.04 | **72.87** +0.51 |

| CodeLlama-7B | MBPP & HumanEval | | | | MBPP$^+$ & HumanEval$^+$ | | | |
|---|---|---|---|---|---|---|---|---|
| | LEDEX SFT | | LEDEX RL | | LEDEX SFT | | LEDEX RL | |
| | Refine | Expl. + Refine | Refine | Expl. + Refine | Refine | Expl. + Refine | Refine | Expl. + Refine |
| pass@1 | 57.21 +14.41 | 56.75 +14.14 | 56.62 -0.60 | **57.41** +0.66 | 49.59 +11.15 | 49.95 +11.55 | 50.73 +1.14 | **51.74** +1.79 |
| pass@10 | 78.17 +5.69 | 78.98 +6.18 | 79.21 +1.04 | **80.25** +1.27 | 72.42 +6.27 | 71.81 +4.74 | 74.03 +1.61 | **74.42** +2.60 |

| CodeLlama-13B | MBPP & HumanEval | | | | MBPP$^+$ & HumanEval$^+$ | | | |
|---|---|---|---|---|---|---|---|---|
| | LEDEX SFT | | LEDEX RL | | LEDEX SFT | | LEDEX RL | |
| | Refine | Expl. + Refine | Refine | Expl. + Refine | Refine | Expl. + Refine | Refine | Expl. + Refine |
| pass@1 | 60.95 +12.23 | 60.64 +11.68 | **61.91** +0.96 | 61.64 +1.00 | 53.86 +11.14 | 53.26 +10.51 | **57.40** +3.54 | 56.60 +3.34 |
| pass@10 | 80.83 +3.69 | 81.29 +3.72 | 81.09 +0.26 | **81.74** +0.45 | 75.21 +4.04 | 74.87 +3.02 | **76.50** +1.29 | 76.00 +1.13 |

RL achieves the highest pass@1 and pass@10 scores (bolded) across all four benchmarks. For CodeLlama-7B, RL achieves the best performance in seven out of eight cases, with SFT yielding the highest pass@1 score on the MBPP benchmark.

For easier comparison, Table 3 summarizes the overall pass@k results on MBPP and HumanEval, along with the improvements achieved through SFT and RL. The improvements from SFT are compared to direct prompting, while the improvements from RL are relative to SFT.

On MBPP and HumanEval overall, SFT boosts StarCoder-15B's pass@1 by 15.34% and pass@10 by 6.58% when directly generating code refinements. When incorporating code explanations in a chain-of-thought format, SFT further enhances StarCoder-15B's performance by 15.92% on pass@1 and 6.30% on pass@10. RL brings an additional 2.43% improvement in pass@1 and 2.55% in

pass@10 for direct refinements, and a further 1.87% pass@1 and 1.85% pass@10 increase when generating both code explanations and refinements. Comparable improvements from SFT and RL are observed across the CodeLlama-7B and CodeLlama-13B models as well.

On the MBPP$^+$ and HumanEval$^+$ benchmarks, which feature more rigorous test cases, respectively [26], we observe even greater improvements from RL training on the CodeLlama models. CodeLlama-7B achieves a 1.79% increase in pass@1 and a 2.60% increase in pass@10 for refined solutions with code explanations. CodeLlama-13B shows a 3.54% improvement in pass@1 and a 1.29% improvement in pass@10 for directly generated refinements. These results demonstrate that RL training enables LLMs to produce or refine solutions that are more robust and capable of passing stricter test cases. Additional experiments and detailed case studies can be found in Appendix A.4.1, A.5.1, A.5.2, and A.5.3.

#### 4.1.2 Success refinement rate

Table 4 presents the refinement success rate for each model backbone across various approaches, averaged over four benchmarks. For StarCoder-15B, the baseline prompting method struggles, achieving only a 6.41% to 6.90% success rate in refining incorrect initial solutions. However, after applying SFT with the high-quality dataset containing code explanations and refinement trajectories, StarCoder-15B demonstrates a notable improvement, raising its refinement success to 16.27% to 16.56%. This increase represents a significant gain of 9.37% to 10.15% over the prompting baseline, showcasing the effectiveness of SFT in enhancing code refinement capabilities by leveraging targeted training data. With further RL, the refinement success for StarCoder-15B improves even more, adding an additional 1.03% to 1.23% over the results from SFT. This final boost highlights the complementary strengths of RL, particularly its capacity to fine-tune model behavior beyond what supervised methods can achieve.

The improvement on CodeLlama-7B and CodeLlama-13B backbones is consistent with that on StarCoder-15B, where RL training eventually achieves the highest success refinement rate with a considerable boost of 1.81 – 3.62%.

Table 4: Success refinement rates over four benchmarks. Blue numbers show the improvement.

| Models | Refine (%) | | | Explain + Refine (%) | | |
| --- | --- | --- | --- | --- | --- | --- |
| | Prompt. | LeDex SFT | LeDex RL | Prompt. | LeDex SFT | LeDex RL |
| StarCoder-15B | 6.90 | 16.27 +9.37 | **17.50** +1.23 | 6.41 | 16.56 +10.15 | **17.59** +1.03 |
| CodeLlama-7B | 8.65 | 18.14 +9.49 | **19.95** +1.81 | 8.10 | 17.60 +9.50 | **20.84** +3.24 |
| CodeLlama-13B | 11.64 | 18.96 +7.32 | **22.58** +3.62 | 11.97 | 20.06 +8.09 | **23.50** +3.44 |

### 4.2 RQ2: Iterative refinement ability

LLMs have the ability to iteratively self-debug until they arrive at correct solutions. Figure 4 illustrates the overall pass@k of CodeLlama-7B across four benchmarks after up to three rounds of refinements. To simplify the figure, we plot the higher pass@k from either the "Refine" or "Expl. + Refine" approach at each refinement round for each model. Additional results on iterative refinement are provided in Appendix A.4.2.

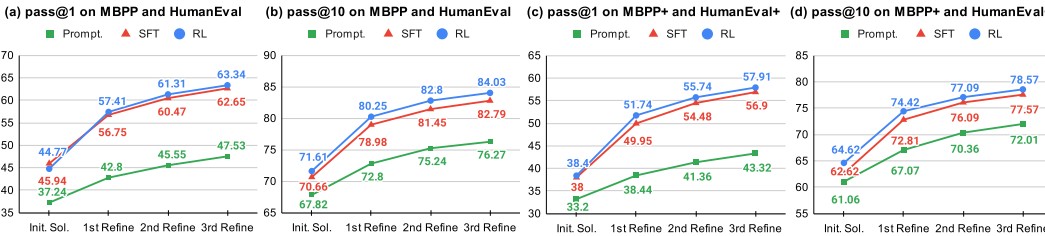

Figure 4: Pass@k of prompting, SFT, and RL CodeLlama-7B after three iterations of refinements.

Both SFT and RL consistently outperform prompting across all three refinement rounds. Even after three rounds, the prompting approach fails to match the pass@k achieved by SFT after just the first

Table 5: Pass@k of CodeLlama-7B trained with CodeLlama-34B's data.

| Models | Approaches | | MBPP | | Humanval | | MBPP$^+$ | | HumanEval$^+$ | |
| --- | --- | --- | --- | --- | --- | --- | --- | --- | --- | --- |
| | | | pass@1 | pass@10 | pass@1 | pass@10 | pass@1 | pass@10 | pass@1 | pass@10 |
| CodeLlama-7B | LEDEX SFT | Init. | 47.01 | 71.18 | 39.24 | 70.00 | 41.64 | 61.95 | 33.07 | 63.58 |
| | | Refine | 54.97 | 76.52 | 47.03 | 77.96 | 48.71 | 68.28 | 41.11 | 71.46 |
| | | Expl. + Refine | 55.05 | 76.63 | 47.34 | 78.11 | 49.15 | 68.35 | 41.49 | 71.76 |
| | LEDEX RL | Init. | 46.45 | 70.41 | 39.63 | 71.02 | 42.34 | 60.84 | 34.15 | 63.19 |
| | | Refine | 54.81 | 76.56 | 48.28 | **80.06** | 53.06 | **70.17** | 47.40 | 73.81 |
| | | Expl. + Refine | **55.32** | **76.74** | **48.52** | 79.14 | **53.17** | 69.36 | **48.59** | **75.80** |

Table 6: Overall pass@k on MBPP & HumanEval and MBPP$^+$ & HumanEval$^+$, trained with CodeLlama-34B's data. Blue numbers show the improvement.

| CodeLlama-7B | MBPP & HumanEval | | | | MBPP$^+$ & HumanEval$^+$ | | | |
| --- | --- | --- | --- | --- | --- | --- | --- | --- |
| | LEDEX SFT | | LEDEX RL | | LEDEX SFT | | LEDEX RL | |
| | Refine | Expl. + Refine | Refine | Expl. + Refine | Refine | Expl. + Refine | Refine | Expl. + Refine |
| pass@1 | 50.01 +7.21 | 53.15 +10.54 | 53.20 +0.05 | **53.64** +0.49 | 45.62 +7.18 | 46.03 +7.63 | 50.76 +5.14 | **51.31** +5.28 |
| pass@10 | 76.88 +4.40 | 77.00 +4.20 | **77.42** +0.42 | 77.33 +0.33 | 69.57 +3.42 | 69.74 +2.67 | 71.65 +2.08 | **71.98** +2.24 |

round (e.g., 47.53% vs. 56.75% in Figure 4 (a)). These results demonstrate that LLMs trained with our pipeline possess strong iterative refinement capabilities, enabling them to achieve progressively higher pass@k with each additional round of refinement.

### 4.3 RQ3: Generalizability of approach

#### 4.3.1 Data collection using open source LLM

To demonstrate the generalizability of LEDEX, particularly the independence of our data collection process from GPT-3.5-Turbo, we substitute GPT-3.5-Turbo with CodeLlama-34B for data collection. As CodeLlama-34B is a pre-trained model, we incorporate few-shot examples in the prompts to guide the generation of incorrect code explanations and refinements. All other processes remain unchanged.

Table 5 presents the pass@k results for CodeLlama-7B trained on data collected from CodeLlama-34B, with Table 6 providing an overall comparison. Although SFT achieves slightly smaller improvements (around 1–3% lower than with GPT-3.5-Turbo data), it still yields notable gains in overall pass@1 and pass@10. Additionally, we observe that RL training further enhances performance on the MBPP$^+$ and HumanEval$^+$ benchmarks, with pass@1 improving by 5.28% and pass@10 by 2.24%. These results demonstrate the generalizability of LEDEX and further suggest that collecting data from a more powerful LLM can lead to better training outcomes within our framework. Additional results can be found in Appendix A.4.3.

#### 4.3.2 Data Collection Using Self-Bootstrap

We also investigate the feasibility of using an LLM to self-bootstrap its training data, specifically by using CodeLlama-7B to generate the data that is then used for its own SFT and RL training.

Table 7 presents the pass@k results of CodeLlama-7B trained with self-bootstrapped data, with Table 8 showing the overall comparison. Compared to prompting, SFT with self-bootstrapped data still delivers up to 8.25% and 2.14% improvements in pass@1 and pass@10 on MBPP and HumanEval, and up to 5.33% and 0.55% improvements on pass@1 and pass@10 on MBPP$^+$ and HumanEval$^+$. Additionally, RL training using the self-bootstrapped data results in a further 0.71% improvement on MBPP and HumanEval, and up to a 0.78% increase on MBPP$^+$ and HumanEval$^+$. These findings suggest that while self-bootstrapped data enables SFT to provide substantial gains over prompting, RL training offers less improvement compared to using data from stronger LLMs, such as CodeLlama-34B or GPT-3.5-Turbo.

### 4.4 RQ4: Quality of generated explanation

We assess whether explanations for incorrect code are useful for developers in understanding their bugs. To do this, we randomly sample 50 problems with initial incorrect solutions from the MBPP and HumanEval benchmarks and use different LLMs to generate explanations for the wrong code.

Table 7: Pass@k of CodeLlama-7B trained with self-bootstrapped data.

| Models | Approaches | | MBPP | | Humanval | | MBPP$^+$ | | HumanEval$^+$ | |
|---|---|---|---|---|---|---|---|---|---|---|
| | | | pass@1 | pass@10 | pass@1 | pass@10 | pass@1 | pass@10 | pass@1 | pass@10 |
| CodeLlama-7B | LEDEX SFT | Init. | 45.83 | 69.24 | 39.85 | 68.83 | 41.78 | 61.77 | 33.25 | 61.50 |
| | | Refine | 52.37 | 74.63 | 47.04 | 74.58 | 46.26 | **66.39** | 40.15 | 67.15 |
| | | Expl. + Refine | 51.80 | **74.99** | 45.70 | 74.72 | 45.94 | 65.77 | 39.10 | 67.33 |
| | LEDEX RL | Init. | 46.28 | 68.87 | 39.90 | 69.49 | 41.61 | 61.29 | 33.66 | 62.17 |
| | | Refine | **52.84** | 74.46 | **48.37** | 75.54 | **46.28** | 65.86 | **41.54** | 68.14 |
| | | Expl. + Refine | 52.34 | 74.60 | 46.90 | **75.70** | 46.10 | 65.99 | 40.79 | **68.50** |

Table 8: Overall pass@k on MBPP & HumanEval and MBPP$^+$ & HumanEval$^+$, trained with self-bootstrapped data. Blue or red numbers show the improvement or deterioration.

| CodeLlama-7B | MBPP & HumanEval | | | | MBPP$^+$ & HumanEval$^+$ | | | |
|---|---|---|---|---|---|---|---|---|
| | LEDEX SFT | | LEDEX RL | | LEDEX SFT | | LEDEX RL | |
| | Refine | Expl. + Refine | Refine | Expl. + Refine | Refine | Expl. + Refine | Refine | Expl. + Refine |
| pass@1 | 51.05 +8.25 | 50.29 +7.68 | **51.74** +0.69 | 51.00 +0.71 | 43.77 +5.33 | 43.16 +4.76 | **44.35** +0.58 | 43.94 +0.78 |
| pass@10 | 74.62 +2.14 | **74.92** +2.12 | 74.73 +0.11 | 74.87 -0.05 | 66.70 +0.55 | 66.40 -0.67 | 66.79 +0.09 | **67.01** +0.61 |

Each explanation is scored on a scale from 1 to 5, based on its correctness and helpfulness, where 1 indicates a completely incorrect or misleading explanation, and 5 denotes a correct explanation that also provides a detailed hint on how to fix the code. Both GPT-4 and human developers are used as evaluators. For GPT-4, we follow prior work [27] and prompt it to score each explanation. The results are presented in Table 9. Both SFT and RL lead to improved explanation quality compared to prompting, with GPT-4 assigning higher scores to models trained using our approach. Notably, the gap between GPT-3.5-Turbo and the trained LLMs significantly narrows after fine-tuning.

Given the time required for human evaluation, we only asked developers to rate explanations from StarCoder models and GPT-3.5-Turbo. Each explanation is scored by two developers, and their ratings are averaged. The human evaluations align with GPT-4's, confirming that SFT improves explanation quality over prompting, while RL further enhances explanations by incorporating code explanation semantics into the reward design. Detailed rubrics and examples of human evaluations can be found in Appendix A.6.

## 5 Limitation

One potential limitation of our study is the reliance on specific large language models (LLMs) from which we collect code explanation and refinement data. Our automated framework is designed to be independent of any specific LLM, and for this study, we use GPT-3.5, CodeLlama-34B, and CodeLlama-7B itself to collect training data, and both bring significant improvement through SFT and RL. However, for future work, it would be interesting to explore the use of other LLMs, including smaller models or a mix of diverse LLMs, to gather explanation and refinement data.

Additionally, our current experiments only use two types of prompts for enabling LLMs to self-debug: one that directly asks for refinement and another that first asks for an explanation of the wrong code followed by refinement. While these prompt designs have shown effectiveness, there might be better prompt strategies for self-debugging that we have not explored due to resource constraints. Exploring a broader range of prompt designs could potentially enhance the performance of our framework. Nonetheless, our proposed training framework is flexible and should be generalizable to different types of data and prompts, paving the way for future innovations in this area.

## 6 Related work

### 6.1 Large language models for code

Large language models (LLMs) have been widely explored across a variety of code-related tasks, including code generation [5, 8, 6, 7, 10, 9, 12, 28–34], bug fixing [35–38], program testing [39, 40] and fuzzing [41] and so on. These models have demonstrated impressive capabilities in these domains, largely due to their strong understanding and generation abilities acquired through extensive pre-

Table 9: Average scores of code explanations rated by GPT-4 and developers. SC for StarCoder and CL for CodeLlama. "-" refers to not applied.

| Raters | Prompt. SC-15B | SFT SC-15B | RL SC-15B | Prompt. CL-7B | SFT CL-7B | RL CL-7B | Prompt. CL-13B | SFT CL-13B | RL CL-13B | Prompt. GPT-3.5-Turbo |
|---|---|---|---|---|---|---|---|---|---|---|
| GPT-4 | 1.90 | 2.92 | 3.18 | 2.00 | 3.00 | 3.10 | 2.54 | 2.62 | 3.24 | 3.60 |
| Developers | 1.73 | 2.60 | 2.76 | - | - | - | - | - | - | 3.21 |

training on vast datasets. This pre-training allows them to recognize patterns, understand context, and generate coherent and contextually relevant code snippets.

However, most existing works in this area focus primarily on improving LLMs to provide the expected output in a single round of generation. The emphasis has been on enhancing the initial output quality, minimizing the need for further modifications or iterations. This one-shot generation approach, while useful, overlooks the potential of iterative refinement, which is a crucial aspect of real-world programming where initial drafts often require multiple rounds of revision and debugging.

### 6.2 Self-debugging and self-refinement

Existing techniques have studied the possibility of using LLMs to refine their generations. Yet, most techniques are prompting LLMs with execution results [13, 14, 42–48] for the refinement. Such prompting approaches bring limited improvement to smaller open-sourced LLMs compared to GPT-3.5. Other techniques train LLMs to self-debug. ILF [49] uses human-annotated feedback information and thus is unscalable, CYCLE [50] and Self-Edit [51] use SFT to fine-tune LLM to generate the refinement only based on the unit test execution feedback. OpenCodeInterpreter [52] and EURUS [53] construct high-quality multi-turn interaction datasets using GPT-3.5-Turbo and GPT-4 to fine-tune LLM for self-refinement.

This work has four differences compared with others that train LLMs: (1) we train LLMs to generate code explanation followed by refinement, which provides additional information to users, (2) we do not require human-annotated training data but propose a scalable pipeline to automatically collect and verify data from another LLM, (3) our data collection pipeline can be generalized to open-sourced LLM or even the same LLM itself, and (4) we design novel reward functions in the RL training stage, considering both the code and explanation quality, which beings extra improvement.

## 7 Conclusion

This work highlights the importance of training open-source LLMs to self-debug and introduces a scalable framework that includes automated data collection, verification, supervised fine-tuning, and reinforcement learning with novel reward designs to enhance LLMs' self-debugging capabilities. Our data collection process is model-agnostic, as demonstrated by the improvements achieved with both GPT-3.5-Turbo and CodeLlama. The data verification ensures high quality of code explanations and refinements. Fine-tuning on this data significantly boosts the LLMs' self-debugging abilities, yielding up to a 15.92% increase in pass@1, a 9.30% increase in pass@10, and a 10.15% increase in successful refinements. Reinforcement learning, utilizing our novel reward design, further enhances performance, with additional gains of up to 3.54% in pass@1, 2.55% in pass@10, and 3.62% in successful refinement rates. Comprehensive analytical experiments confirm the generalizability of our approach and demonstrate the iterative refinement capabilities of the trained models. Moreover, human evaluations indicate that the LLMs trained with our framework produce higher-quality explanations, effectively aiding developers in understanding and resolving bugs in source code.

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

# A   Appendix

## A.1   Limited refinement ability of open-sourced LLMs

We evaluate two prompting strategies for querying LLMs to perform code refinement: one where the LLMs directly generate the refinement (Figure 5 (3.1)), and another where they first explain why the code is incorrect before generating the refinement (Figure 5 (3.2)). Table 10 reports the success rates of refinements generated by StarCoder-15B, CodeLlama-7B, CodeLlama-13B, and GPT-3.5 using greedy decoding. Generally, LLMs that first explain the incorrect code are more likely to generate accurate refinements. However, despite these efforts, all three open-source LLMs exhibit limited refinement capabilities. For example, StarCoder-15B successfully refines only 4.43% to 5.10% of incorrect code, underscoring the need for training open-source LLMs to enhance their ability to explain and refine code.

Table 10: The success rate of self-refinement using greedy decoding.

| Models | MBPP | | HumanEval | |
|---|---|---|---|---|
| | Refine | Expl. + Refine | Refine | Expl. + Refine |
| StarCoder-15B | 2.58 | 4.43 | 2.04 | 5.10 |
| CodeLlama-7B | 7.42 | 6.71 | 5.32 | 8.52 |
| CodeLlama-13B | 9.80 | 10.20 | 5.88 | 8.24 |
| GPT-3.5 | 26.24 | 28.90 | 32.32 | 33.33 |

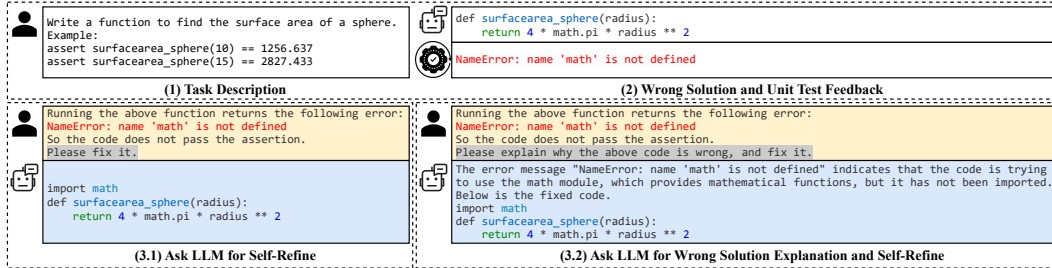

Figure 5: Two different prompts to ask LLM to self-refine: directly asking for refinement (left), asking for an explanation of the wrong code, and then refining in chain-of-thought (right).

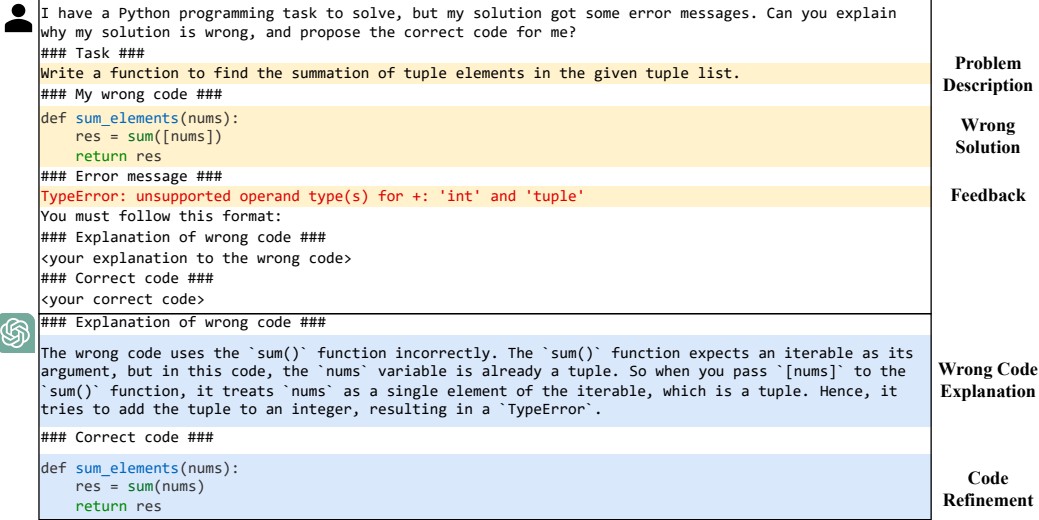

Figure 6: Prompt used for code explanation and refinement data collection.

## A.2 Data collection and data format for training

Figure 6 provides an example of the prompt we used to collect code explanation and refinement training data from GPT-3.5 (the same approach was used for CodeLlama-34B). We set the temperature to 0.8, allowing GPT-3.5 to generate 10 code explanations and refinements per prompt. The collected data includes both the explanation of the incorrect code and the corresponding refinement. From this, we construct two formats of instruction-tuning data for supervised fine-tuning (SFT).

Figure 7 (left) shows an instruction that asks LLMs to provide only the code refinement, where the instruction includes the task description, the incorrect initial solution, and execution feedback. The LLMs are trained to generate the refined solution based on this input. Figure 7 (right) presents a different instruction that asks for both the code explanation and the refinement. By incorporating both types of instruction data during training, we increase the diversity of the training set and improve the LLMs' robustness across different prompts.

## A.3 PPO algorithm with separate rewards in RL

We modify the standard PPO algorithm to optimize the explanation and refinement generation based on their reward separately. Given an LLM (already supervise-fine-tuned) $\pi$ as the policy model, a prompt $x$, a generation $y$ consists of a code explanation and a refinement:

$$y = [e, r] = \{y_1, y_2, \ldots, y_{|e|}, y_{|e|+1}, y_{|e|+2}, \cdots, y_{|e|+|r|}\}$$

$\{y_i\}_{i=1}^{|e|}$ is the explanation of the wrong code, and $\{y_i\}_{i=|e|}^{|e|+|r|}$ is the refinement. We then define the advantage $A$ in the PPO algorithm as:

Figure 7: Two types of instruction tuning data used in SFT and RL.

$$A_t = \delta_t + \gamma\delta_{t+1} + \ldots + \gamma^{\mathcal{T}-t}\delta_{\mathcal{T}} \quad , \quad \delta_t = \mathcal{R}_t - V(y_{<t}, x, \pi) + \gamma V(y_{<t+1}, x, \pi)$$

$A_t$ is the advantage at decoding timestamp $t$, $\mathcal{T} = |e| + |r|$ is the total length of generation output, and $\gamma$ is the discount rate (a hyper-parameter set to 0.99 in our experiment). $V(y_{<t}, x, \pi)$ is the state value at generation step $t$ given input $x$, which is learned and calculated by a linear layer on top of the policy model $\pi$. $\mathcal{R}_t$ is the reward at decoding timestamp $t$, which is calculated as follow:

$$\mathcal{R}_t = \begin{cases} \mathcal{R}(r) - \mathrm{KL}_t(\pi, \pi') & , t = \mathcal{T} \\ \mathcal{R}(e) - \mathrm{KL}_t(\pi, \pi') & , t = |e| \\ \mathrm{KL}_t(\pi, \pi') \approx \log \dfrac{\mathrm{P}(y_t|y_{t-1}, x, \pi)}{\mathrm{P}(y_t|y_{t-1}, x, \pi')} & , \text{otherwise} \end{cases}$$

where KL is the Kullback–Leibler divergence [54] between the action distribution given by the updated policy model $\pi$ and the old policy model $\pi'$ before the update.

Instead of assigning both the code and explanation rewards to the entire output, we separate them by only assigning the explanation reward to the explanation portion. This avoids the issue that a low reward is assigned to a correct explanation followed by a wrong refinement, and the LLM learns to keep the correctly generated explanation part and focus on improving the incorrect refinement portion. For data where only code refinement but no explanation is generated, we use the standard design to assign the code reward to the code refinement.

Following existing works, the loss of the PPO algorithm training is:

$$L = -\mathbb{E}\left[\sum_{t=1}^{\mathcal{T}} \frac{\log \mathrm{P}(y_t|y_{<t}, x, \pi)}{\log \mathrm{P}(y_t|y_{<t}, x, \pi')} A^t\right] + \alpha\mathbb{E}\left[\sum_{t=1}^{\mathcal{T}} \left(V(y_{<t}, x, \pi) - \left(A_t + V(y_{<t}, x, \pi')\right)\right)^2\right]$$

By minimizing this loss, the policy model $\pi$ (the LLM under PPO training) is trained to generate explanations and refinements with higher reward but also constrained by not being distracted too much far away from the supervised-fine-tuned LLM $\pi'$.

## A.4 Additional results

### A.4.1 Comparison with SFT on code generation

Table 11 shows the comparison between CodeLlama-7B trained with LEDEX, and the CodeLlama-7B trained with code generation data only. Although training with code generation enables LLMs to get comparable (or even higher pass@k on HumanEval) pass@k on one round of code generation, the

LLMs cannot obtain strong self-debugging ability from code generation data. The LLM trained with code generation data can only improve the pass@k very little after self-debugging. By contrast, the LLM trained with the full collected data (code generation, code explanation, and code refinement data) gets significantly higher pass@k after refinement, showing its strong self-debugging ability.

Table 11: Pass@k on MBPP and HumanEval by CodeLlama-7B trained on code generation only, and that trained with our collected code explanation and refinement data.

| Models | Approaches | MBPP | | Humanval | | MBPP$^+$ | | HumanEval$^+$ | |
|---|---|---|---|---|---|---|---|---|---|
| | | pass@1 | pass@10 | pass@1 | pass@10 | pass@1 | pass@10 | pass@1 | pass@10 |
| Only Code Generation Data | Init. | 46.18 | 70.56 | 40.14 | 70.37 | 41.23 | 60.58 | 34.22 | 63.34 |
| | Refine | 49.18 | 72.78 | 42.91 | 74.57 | 43.13 | 61.81 | 36.12 | 65.77 |
| | Expl. + Refine | 49.58 | 73.07 | 43.88 | 75.19 | 43.58 | 61.69 | 36.37 | 66.24 |
| LEDEX's Full Data | Init. | 48.87 | 70.89 | 36.99 | 69.95 | 42.97 | 62.69 | 30.76 | 62.52 |
| | Refine | 58.07 | 77.34 | 52.65 | 80.71 | 51.64 | 71.04 | 46.61 | 74.43 |
| | Expl. + Refine | 57.98 | 77.92 | 52.98 | 82.22 | 51.55 | 70.94 | 47.62 | 75.54 |

### A.4.2    Iterative refinement

Figure 8 illustrates the pass@k metric when employing trained CodeLlama-13B for iterative code refinement. The findings align with those depicted in Figure 4, that both SFT and RL consistently outperform the prompting approach in each round of refinement. Notably, even after three rounds of refinement, the prompting method fails to achieve a higher pass@k than what SFT and RL models attain after just one round. This highlights the substantial advantage of SFT and RL methods in enhancing code quality. Moreover, the SFT and RL CodeLlama-13B models demonstrate robust continuous refinement capabilities, maintaining their superior performance across multiple iterations. This consistent outperformance underscores the effectiveness of SFT and RL strategies in refining and improving code, suggesting their potential for more efficient and reliable coding practices in iterative development scenarios.

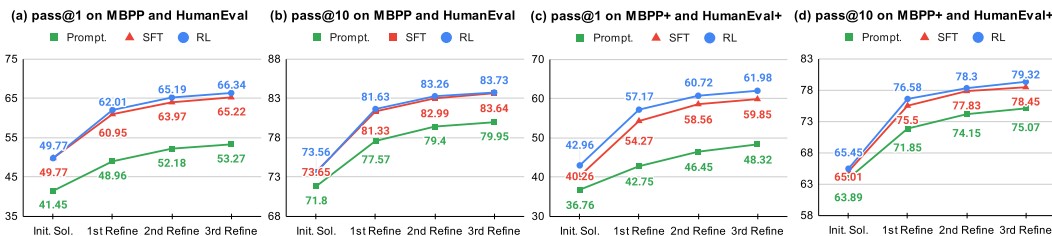

Figure 8: Pass@k of prompting, SFT, and RL CodeLlama-13B after three iterations of refinements.

### A.4.3    Generalizability

The pass@k of CodeLlama-13B trained with data collected from CodeLlama-34B are shown in Table 12, where RL achieves the highest pass@1 and pass@10 in all the benchmarks. The overall results are shown in Table 13, and SFT improves the pass@1 by up to 9.14% and pass@10 by 3.47% on the overall of MBPP and HumanEval, and improves the pass@1 by up to 8.68% and pass@10 by 3.92% on MBPP$^+$ and HumanEval$^+$. Further RL training improves the pass@1 by up to 0.61% and pass@10 by 0.86%. The improvement brought by RL is significantly larger on MBPP$^+$ and HumanEval$^+$ (up to 6.80% higher pass@1 and 2.28% higher pass@10), which is consistent with our finding when using GPT-3.5-Turbo's training data that RL brings more considerable improvement on harder benchmarks.

Table 14 presents the success refinement rates achieved by CodeLlama-13B when trained on data collected from CodeLlama-34B, averaged across four benchmarks. SFT refines 15.60% to 16.83% of incorrect solutions, outperforming the prompting approach by 3.96% to 4.86%. RL training further boosts the refinement rate, improving it by 4.78% to 7.14% over SFT. Notably, the improvement from SFT is somewhat smaller when using CodeLlama-34B's data compared to GPT-3.5-Turbo's data, likely due to the slightly lower quality of explanations and refinements generated by CodeLlama-34B. However, RL training raises the refinement rate to levels comparable to those achieved with GPT-3.5-

Table 12: Pass@k on MBPP and HumanEval by LLMs trained with CodeLlama-34B's data.

| Models | Approaches | | MBPP pass@1 | MBPP pass@10 | Humanval pass@1 | Humanval pass@10 | MBPP$^+$ pass@1 | MBPP$^+$ pass@10 | HumanEval$^+$ pass@1 | HumanEval$^+$ pass@10 |
|---|---|---|---|---|---|---|---|---|---|---|
| CodeLlama-13B | Prompt. | Init. | 42.88 | 70.85 | 37.11 | 74.69 | 38.93 | 62.26 | 30.15 | 66.27 |
| | | Refine | 49.68 | 75.85 | 45.78 | 81.07 | 46.22 | 70.14 | 37.62 | 72.68 |
| | | Expl. + Refine | 49.97 | 76.39 | 45.90 | 81.18 | 45.77 | 70.48 | 38.36 | 73.84 |
| | LEDEX SFT | Init. | 50.61 | 73.14 | 44.96 | 76.85 | 44.66 | 63.15 | 37.98 | 69.59 |
| | | Refine | 59.07 | 79.52 | 54.15 | 83.95 | 52.87 | 73.50 | 46.96 | 77.40 |
| | | Expl. + Refine | 59.11 | 79.56 | 55.04 | 84.75 | 53.97 | 73.66 | 47.73 | 78.17 |
| | LEDEX RL | Init. | 50.50 | 73.26 | 45.62 | 78.14 | 44.72 | 63.40 | 38.79 | 71.98 |
| | | Refine | 59.15 | **80.21** | 55.71 | 85.33 | **57.74** | **74.03** | **56.56** | 82.23 |
| | | Expl. + Refine | **59.57** | 80.18 | **56.08** | **85.40** | 56.60 | 73.17 | 56.24 | **82.39** |

Table 13: Overall pass@k on MBPP & HumanEval and MBPP$^+$ & HumanEval$^+$, trained with CodeLlama-34B's data. Blue numbers show the improvement.

| CodeLlama-13B | MBPP & HumanEval | | | | MBPP$^+$ & HumanEval$^+$ | | | |
|---|---|---|---|---|---|---|---|---|
| | LEDEX SFT | | LEDEX RL | | LEDEX SFT | | LEDEX RL | |
| | Refine | Expl. + Refine | Refine | Expl. + Refine | Refine | Expl. + Refine | Refine | Expl. + Refine |
| pass@1 | 57.85 +9.13 | 58.10 +9.14 | 58.30 +0.45 | **58.71** +0.61 | 50.46 +7.74 | 51.43 +8.68 | **57.26** +6.80 | 56.45 +5.02 |
| pass@10 | 80.61 +3.47 | 80.84 +3.27 | **81.47** +0.86 | **81.47** +0.63 | 75.09 +3.92 | 75.5 +3.65 | **77.37** +2.28 | 76.92 +1.42 |

Turbo's data, indicating that RL training can mitigate the quality differences between data generated by open-source LLMs and commercial LLMs.

## A.5 Case studies

### A.5.1 Correct refinements generated by SFT LLMs

Figure 9 shows an example from the HumanEval benchmark for which the prompting approach fails to generate correct refinement. The bug in the initial solution is that 'nlargest' returns the largest elements in descending order, but from the example provided, one can find out that the expected output is in ascending order. The prompting approach does not work for this example, and the prompted StarCoder generates a hallucinated explanation and simply repeats the wrong solution. Actually, such simple repeats of wrong solutions are very common when using the prompting approach, which supports our motivation to train LLM to self-debug. However, after SFT training, the trained StarCoder can correctly figure out the reason for test failure and generate the correct refinement.

### A.5.2 Correct refinements only generated by RL LLMs

Figure 10 shows an example from the MBPP benchmark for which only the StarCoder trained with reinforcement learning explains the wrong code correctly, pointing out that the failing test case is because "this approach does not consider the case where multiple elements have the same maximum count" (highlighted with green background), and generates the correct refinement. The SFT StarCoder is unable to diagnose the wrong code correctly.

### A.5.3 Robust refinements generated by RL LLMs

Figure 11 shows an example from HumanEval, for which both SFT and RL CodeLlama-13B generate the correct refinements that pass all the test cases from the HumanEval benchmark. Yet, the refinement from CodeLlama-13B is not fully correct, as "(x, y, z) == ((int)x, (int)y, (int)z)" is not equivalent to checking if these numbers are integers. The refinement generated by SFT CodeLLama-13B (left) fails the harder test cases in HumanEval$^+$, i.e., AssertionError: expect any_int(3.0,

Table 14: Success refinement rate of different approaches over four benchmarks, trained with CodeLlama-34B's data. Blue numbers show the improvement.

| Models | | Refine (%) | | | Explain + Refine (%) | |
|---|---|---|---|---|---|---|
| | Prompt. | LEDEX SFT | LEDEX RL | Prompt. | LEDEX SFT | LEDEX RL |
| CodeLlama-13B | 11.64 | 15.60 +3.96 | **22.74** +7.14 | 11.97 | 16.83 +4.86 | **21.61** +4.78 |

```
<|user|>Given an array arr of integers and a positive integer
k, return a sorted list of length k with the maximum k numbers
in arr.
Example 1:
Input: arr = [-3, -4, 5], k = 3
Output: [-4, -3, 5]
...
Note:
1. The length of the array will be in the range of [1, 1000].
2. The elements in the array will be in the range of [-1000,
1000].
3. 0 <= k <= len(arr)
<|end|>
<|assistant|>
def maximum(arr, k):
    from heapq import nlargest
    return nlargest(k, arr)
<|end|>
<|user|> Running the above code returns the following error
message:
AssertionError: expect maximum([-3, -4, 5], 3) == [-4, -3, 5],
but was [5, -3, -4]
So the code does not pass the assertion. Please:
1. Explain why the above code is wrong.
2. Propose the correct code.
<|end|>
```

```
<|assistant|>
The provided code for the maximum function uses the heapq
module to find the k largest elements in the array.
However, the heapq module only supports finding the k largest
elements in an array.
It doesn't support finding the k smallest elements in an
array.
To fix this issue, you need to use the nsmallest function from
the heapq module.
def maximum(arr, k):
    from heapq import nlargest
    return nlargest(k, arr)
<|end|>                                                        ❌
```

**Incorrect Explanation and Refinement Generated by Prompting StarCoder**

```
<|user|>Given an array arr of integers and a positive integer
k, return a sorted list of length k with the maximum k numbers
in arr.
Example 1:
Input: arr = [-3, -4, 5], k = 3
Output: [-4, -3, 5]
...
Note:
1. The length of the array will be in the range of [1, 1000].
2. The elements in the array will be in the range of [-1000,
1000].
3. 0 <= k <= len(arr)
<|end|>
<|assistant|>
def maximum(arr, k):
    from heapq import nlargest
    return nlargest(k, arr)
<|end|>
<|user|> Running the above code returns the following error
message:
AssertionError: expect maximum([-3, -4, 5], 3) == [-4, -3, 5],
but was [5, -3, -4]
So the code does not pass the assertion. Please:
1. Explain why the above code is wrong.
2. Propose the correct code.
<|end|>
```

```
<|assistant|>
The wrong code is using the `nlargest` function from the
`heapq` module to find the k largest elements in the array.
However, the `nlargest` function returns the largest elements
in descending order, while we need the largest elements in
ascending order. This is why the output is reversed.

def maximum(arr, k):
    from heapq import nlargest
    return nlargest(k, arr)[::-1]
<|end|>                                                        ✅
```

**Correct Explanation and Refinement Generated by SFT StarCoder**

Figure 9: Example for which prompting StarCoder fails to but SFT StarCoder generates the correct explanation and refinement.

4, 7) == False, but was True, since it fails to realize that 3.0 is not integer. By contrast, the refinement generated by the RL CodeLlama-13B generates better refinement that passes all the test cases from HumanEval$^+$.

## A.6  Human rating of code explanation

We let developers rate the explanations generated by different LLMs based on the following rubrics, which consider both the correctness of the statements in the explanations and also helpfulness of the explanation.

Table 15: Rubrics used for developer rating of LLMs' generated explanations.

| Rates | 1 | 2 | 3 | 4 | 5 |
|---|---|---|---|---|---|
| Correctness | Totally wrong or misleading | Mostly wrong but with at least one minor correct point | Partially correct explanation with about 50% wrong | Partially correct explanation with minor mistakes | Totally correct explanation |
| Helpfulness | Helpless statement that repeats the error or is not related to the bug | Not very helpful but at least related to the bug | Provides very vague hints about how to understand the failed test case | Provides vague hints about how to fix the wrong code | Provides at least one clear hints about how to fix |

Figure 12 shows an example of developer rating, where the prompting and SFT StarCoder are rated "1" since their explanation is wrong and not informative. yet, the RL StarCoder's explanation successfully points out that the function should return 'Not Matched!' when there is no match.

Figure 13 shows another example, where the prompting StarCoder is rated "2", as the developer thinks the explanation is stating a correct fact. By contrast, SFT and RL StarCoder correctly point out that the function should check if the array is empty or not.

<|user|> Write a python function to find the item with maximum occurrences in a given list.
Example: assert max_occurrences([1,2,3,1,2,3,12,4,2]) == 2
<|end|>
<|assistant|>
```python
def max_occurrences(nums):
    result = max(set(nums), key = nums.count)
    return result
```
<|end|>
<|user|> Running the above code returns the following error message:
AssertionError: expect max_occurrences([1,2,6,7,0,1,0,1,0]) == (1,0), but was 0
So the code does not pass the assertion. Please:
1. Explain why the above code is wrong.
2. Propose the correct code.
<|end|>

<|assistant|>
The wrong code is using the `max` function to find the item with the maximum occurrences in the dictionary `d`. However, the `max` function returns the key with the maximum value, not the key-value pair itself. So, the code is returning just the key instead of the key-value pair.
```python
def max_occurrences(numbers):
    d = {}
    for n in numbers:
        if n in d:
            d[n] += 1
        else:
            d[n] = 1
    return max(d.items(), key=lambda x: x[1])
```
<|end|>

**Incorrect Explanation and Refinement Generated by SFT StarCoder**

---

<|user|> Write a python function to find the item with maximum occurrences in a given list.
Example: assert max_occurrences([1,2,3,1,2,3,12,4,2]) == 2
<|end|>
<|assistant|>
```python
def max_occurrences(nums):
    result = max(set(nums), key = nums.count)
    return result
```
<|end|>
<|user|> Running the above code returns the following error message:
AssertionError: expect max_occurrences([1,2,6,7,0,1,0,1,0]) == (1,0), but was 0
So the code does not pass the assertion. Please:
1. Explain why the above code is wrong.
2. Propose the correct code.
<|end|>

<|assistant|>
The wrong code uses the `max()` function with the `key` parameter set to `items.count`. This means that it will find the element in the set `items` that has the maximum count in the original list. However, this approach does not consider the case where multiple elements have the same maximum count. In such cases, the `max()` function will return the first element with the maximum count, which may not be the correct answer.
```python
def max_occurrences(numbers):
    d = {}
    for n in numbers:
        if n in d:
            d[n] += 1
        else:
            d[n] = 1
    max_count = max(d.values())
    max_nums = [num for num, count in count_dict.items()
        if count == max_count]
    if len(max_nums) == 1:
        return max_nums[0]
    else:
        return tuple(max_nums)
```
<|end|>

**Correct Explanation and Refinement Generated by RL StarCoder**

Figure 10: Example for which the RL LLM generates the correct explanation and refinement.

---

<|user|>Create a function that takes 3 numbers.
Returns true if one of the numbers is equal to the sum of the other two, and all numbers are integers. Returns false in any other cases.
Examples
any_int(5, 2, 7) → True
any_int(3, 2, 2) → False
any_int(3, -2, 1) → True
any_int(3.6, -2.2, 2) → False
<|end|>
<|assistant|>
```python
def any_int(x, y, z):
    if x + y == z or x + z == y or y + z == x:
        return True
    return False
```
<|end|>
<|user|> Running the above code returns the following error message:
AssertionError: expect any_int(1.5, 5, 3.5) == False, but was True
So the code does not pass the assertion. Please:
1. Explain why the above code is wrong.
2. Propose the correct code.
<|end|>

<|assistant|>
The wrong code is not checking if all the numbers are integers. It only checks if the sum of two numbers is equal to the third number. However, the task requires checking if all three numbers are integers.
```python
def any_int(x, y, z):
    if (x, y, z) == (int(x), int(y), int(z)):
        if x + y == z or x + z == y or y + z == x:
            return True
    return False
```
<|end|>

**Non-robust Refinement Generated by Prompting SFT CodeLlama-13B**

---

<|user|>Create a function that takes 3 numbers.
Returns true if one of the numbers is equal to the sum of the other two, and all numbers are integers. Returns false in any other cases.
Examples
any_int(5, 2, 7) → True
any_int(3, 2, 2) → False
any_int(3, -2, 1) → True
any_int(3.6, -2.2, 2) → False
<|end|>
<|assistant|>
```python
def any_int(x, y, z):
    if x + y == z or x + z == y or y + z == x:
        return True
    return False
```
<|end|>
<|user|> Running the above code returns the following error message:
AssertionError: expect any_int(1.5, 5, 3.5) == False, but was True
So the code does not pass the assertion. Please:
1. Explain why the above code is wrong.
2. Propose the correct code.
<|end|>

<|assistant|>
The wrong code does not check if all the numbers are integers. It only checks if one of the numbers is equal to the sum of the other two. This is why the assertion `any_int(1.5, 5, 3.5)` returns True, which is incorrect.
```python
def any_int(x, y, z):
    if x + y == z or x + z == y or y + z == x:
        if all(isinstance(num, int) for num in [x, y, z]):
            return True
    return False
```
<|end|>

**Robust Refinement Generated by RL CodeLlama-13B**

Figure 11: Example for which RL LLM generates more robust refinement.

```
<|user|>Write a function to search some literals strings in
a string by using regex.
Example:
assert check_literals('The quick brown fox jumps over the
lazy dog.', ['fox']) == 'Matched!'
<|end|>

<|assistant|>
import re
def check_literals(text,literals):
    for literal in literals:
        if re.search(literal,text):
            return "Matched!"
    return "Not matched!"
<|end|>
<|user|> Running the above code returns the following error
message:
AssertionError: expect check_literals('The quick brown fox
jumps over the lazy dog.',['horse']) == 'Not Matched!', but
was 'Not matched!'
So the code does not pass the assertion. Please:
1. Explain why the above code is wrong.
2. Propose the correct code.
<|end|>
```
**Task Description, Wrong Solution and Execution Feedback**

```
"AssertionError: expect check_literals('The quick brown
fox jumps over the lazy dog.',['horse']) == 'Not
Matched!', but was 'Not matched!'" indicates that the
code is trying to check if the string 'horse' is in the
text, but the string 'horse' is not in the text.
```
**Explanation by Prompting StarCoder (Rating: 1)**

```
The wrong code is not handling the case when there is no
match for any of the literals. It is returning 'Not
matched!' immediately when there is no match, instead of
checking all the literals.
```
**Explanation by SFT StarCoder (Rating: 1)**

```
The wrong code is not handling the case when there is no
match for any of the literals. It is returning 'Not
matched!' in such cases, but the assertion expects 'Not
Matched!'. This is causing the assertion error.
```
**Explanation by RL StarCoder (Rating: 4)**

Figure 12: Example of human rating, where RL StarCoder generates the best explanation.

```
<|user|>Given an array of non-negative integers, return a
copy of the given array after sorting,
you will sort the given array in ascending order if the sum(
first index value, last index value) is odd,
or sort it in descending order if the sum( first index
value, last index value) is even.
Examples:
* sort_array([]) => []
* sort_array([5]) => [5]
* sort_array([2, 4, 3, 0, 1, 5]) => [0, 1, 2, 3, 4, 5]
<|end|>

<|assistant|>
def sort_array(array):
    if sum(array[0], array[-1]) % 2 == 0:
        return sorted(array, reverse=True)
    else:
        return sorted(array)
<|end|>
<|user|> Running the above code returns the following error
message:
IndexError: list index out of range
So the code does not pass the assertion. Please:
1. Explain why the above code is wrong.
2. Propose the correct code.
<|end|>
```
**Task Description, Wrong Solution and Execution Feedback**

```
The error message "IndexError: list index out of range"
indicates that the code is trying to access an element in
the array that does not exist.
```
**Explanation by Prompting StarCoder (Rating: 2)**

```
The error message "IndexError: list index out of range"
is occurring because the code is trying to access the
first and last elements of the array using array[0] and
array[-1], but if the array is empty, it will result in
an index out of range error.
```
**Explanation by SFT StarCoder (Rating: 4)**

```
The error message "IndexError: list index out of range"
occurs because the code is trying to access the first and
last elements of the array using `array[0]` and
`array[-1]`, but if the array is empty, it will result in
an index out of range error.
```
**Explanation by RL StarCoder (Rating: 4)**

Figure 13: Example of human rating, where both SFT and RL StarCoder generate good explanations.

## A.7 Potential impact

Using LLMs to help coding is popular nowadays. This work proposes a technique to train LLMs to explain and self-refine code, which aims to improve developers' coding experience with LLMs. We also call for training LLMs to take feedback beyond the prompting approach to improve the LLMs' self-debugging ability. The technique is supposed to be on the same track as all the existing LLMs for code generation. Thus, we think no special concerns about broader impact need to be highlighted here.

