# OpenReview forum: "LeDex: Training LLMs to Better Self-Debug and Explain Code"
_NeurIPS.cc/2024/Conference — NeurIPS 2024 poster_

### Official Review · Reviewer_osNA · 2024-07-03

**Soundness:** 3
**Presentation:** 3
**Contribution:** 2
**Rating:** 7
**Confidence:** 4

**Summary:**

The paper addresses the goal of self-debugging of generated code, while also explaining it.
The approach is to: (a) sample code outputs to natural language inputs, and keep only the wrong code outputs according to unit tests; (b) sample **refinements** ("fixes") to the wrong code outputs and test those refinements using unit tests, to get both correct and incorrect refinements; (c) the authors train an LLM using SFT and RL on those correct and incorrect refinements.

The resulting trained LLM is shown to provide empirical gains, especially when allowing it to self-refine its outputs at test time, with the test-time unit tests.

**Strengths:**

* The proposed pipeline provide strong empirical gains
* The authors experiment with 3 base LLMs (StarCoder-15B, CodeLlama-7B, CodeLlama 13B) and 2 "teacher" LLMs (gpt-3.5-turbo, CodeLlama-34B)
* The authors experiment with multiple datasets: they train on MBPP, APPS, CodeContest, and test on HumanEval and MBPP.

**Weaknesses:**

* The paper is mostly applicative, and includes a mix of known techniques. I feel that the paper is low on conceptual novelty. The concept of self-debugging and verifying correctness using unit tests was introduced by [Chen et al., 2023](https://arxiv.org/pdf/2304.05128) (although only using prompting), and RL using the signal coming from the unit tests was done in several papers (that the authors cite).
* It feels like there are so many different techniques and heuristics involved, that it is hard to pinpoint the exact contribution of each of them. For example: the design of the explanation score as $CosSim(RoBERTa(e), RoBERTa(ec))$, the exact hyperparameters of the reward design (e.g., $\frac{50 · S_{ex}(e) 35}{3}$), and the exact design of $S_{cb}$ using CodeBLEU. The use of explanations before fixing the code is a form of Chain-of-Thought (Wei et al., 2022), or the "feedback" in Self-Refine (Madaan et al., 2023).
* Further, teaching the models to self-debug is done using **larger models**.
That is, the refinements are sampled from larger models such as gpt-3.5-turbo, CodeLlama-34B, and then these refinements are used to train the smaller StarCoder-15B, CodeLlama-7B, CodeLlama 13B models.
This adds a dimension of distillation (from large models to small models), and further makes it difficult to pinpoint the exact source of contribution.
* The approach is not compared to any related work, or to the teacher models themselves.
* Position in literature - Although many papers are cited, I feel that the paper does not position itself well in the literature. I do not remember exactly what did each related work do, but the paper does not help me understand the differences and its novelty compared to the related work. For example, what's the difference between this paper and [18] and [20]? They seem very similar, but the Related Work does not highlight the novelty over them.
    * Another new paper that was not cited: [Ansong et al., NExT: Teaching Large Language Models to Reason about Code Execution, ICML'2024](https://arxiv.org/pdf/2404.14662). I am not interested in the authors just citing the paper as an additional number between brackets, I am interested in discussing the actual differences and novelty over that paper.

**Questions:**

1. At test time, when evaluating the SFT/RL models: does the model see the execution results of its generated test code before refinement? That is, can the model use the unit tests of the test examples as well, or are unit tests used only at training time?
2. In the definition of $S_{cb}$ - if we rely on unit tests to verify correctness, why do we need to encourage the refinement to be similar (in terms of CodeBLEU) to **all** the possible correct refinements?
3. RL training on benchmark data may over-specialize on their specific domain, while degrading the general coding abilities of the LLM. After all, these benchmarks are only benchmarks, and over-specializing on them may hurt the usability of the model in practical use. For example, have the authors checked whether applying their SFT+RL hurts the perplexity on general code?

To summarize, I think that the paper presents strong empirical gains, but the scientific novelty is low, as the contribution is mostly applied. I thus vote for a borderline reject.

---

> ### Author Rebuttal · Authors · 2024-08-07
>
> We thank the reviewer for insightful suggestions and questions.
>
> ## 1. Paper novelty
> While there are several related works on self-debugging, our paper focuses on how to improve the model’s self-debugging capability, which is important but not yet extensively investigated.
>
> We believe "NExT: Teaching Large Language Models to Reason about Code Execution" is a concurrent work with us. Yet there are still differences between the NExT paper and our paper:
> 1. One of the most notable differences is the proposed **RL training with explanation and execution reward**. RL training is important as it helps LLMs to learn from the failure generations as well, and the separation of explanation reward and execution reward helps LLMs learn differently about the explanations and fixes.
> 2. Differences also lie in how we synthesize explanations while they focus on traces reasoning, and how we **utilize much larger training datasets such as APPS and CodeContests** to improve generalization while they primarily focus on MBPP and HumanEval as the training data.
> 3. Also, they mainly conduct training experiments on PaLM 2, while we **train multiple open-sourced backbones to prove the generalizability of our approach**.
>
> We will add a discussion on the differences with this paper in our draft, but we do not think that the contribution and novelty of our paper should be questioned based on this concurrent work.
>
> ## 2. Position in literature
> As mentioned above, our paper focuses on how to improve the model’s self-debugging capability via training. Most existing works on self-debugging focus on prompting LLMs to do self-debugging, which does not work well on open-sourced smaller LLMs as we have shown.
>
> A few related works that train LLMs as we cited in the paper:
> 1. ILF requires human-annotated explanations, while we train LLMs to generate bug explanations by themselves.
> 2. CYCLE and Self-Edit only train LLMs to generate refinement using SFT. We train LLMs to explain the bug which not only enhances LLMs’ reasoning but also helps developers to understand the wrong code (as our human evaluation shows). We also explored using RL to further improve the self-debugging performance.
> 3. NExT, a concurrent work from ICML this year that we have discussed above.
>
> We are also different from all these works by designing RL training. The RL training brings improvement to the strong SFT models over several baselines, leading to higher Pass@K, higher refinement rate, and better bug explanation.
>
> ## 3. Self-taught refinement
> We provide the experiment results using the synthetic data generation from the model itself for CodeLlama 7B. We highlight some results here. Below is the CodeLlama-7B SFT/RL using the data collected from itself, evaluated on MBPP+ and HumanEval+.
>
> | Approaches | | MBPP+ | | HumanEval+ | |
> |-|-|-|-|-|-|
> | | | Pass@1 | Pass@10 | Pass@1 | Pass@10 |
> | | Init. | 37.18 | 61.23 | 27.40 | 60.81 |
> | Prompt | Refine | 42.97 | 66.89 | 31.84 | 65.08 |
> | | Expl. + Ref. | 42.46 | 67.41 | 32.49 | 66.58 |
> |||||||
> | | Init. | 41.78 | 61.77 | 33.25 | 61.50 |
> | SFT | Refine | 46.26| **66.39** | 40.15 | 67.15 |
> | | Expl. + Ref. | 45.94 | 65.77 | 39.10 | 67.33 |
> |||||||
> | | Init. | 41.61 | 61.29 | 33.66 | 62.17 |
> | RL | Refine | **46.28** | 65.86 | **41.54** | 68.14 |
> | | Expl. + Ref. | 46.10 | 65.99 | 40.79 | **68.50** |
>
> The full results are in our attached author response PDF file, Tables 1 and 2. Results show that self-taught SFT and RL also achieve big improvements. CodeLlama-7B SFT/RL models achieve up to 5% improvement in self-debugging, compared with the baseline prompting method, and the model trained with code generation data only. Though comparing the experiments using data from CodeLlama-34B and GPT-3.5-Turbo, the improvement is less.
>
> ## 4. Comparison to the teacher models
> We provide the comparison with the teacher models CodeLlama-34B and GPT-3.5-Turbo on the self-debugging setup in the global rebuttal pdf. Comparing it with Table 2 and Table 10 in the paper, we see that with CodeLlama-34B as the teacher, CodeLlama 7B SFT/RL achieves close to CodeLlama-34B self-debugging performance, while CodeLlama-13B SFT/RL significantly outperforms the CodeLlama-34B teacher (e.g. in HumanEval+ pass@1 56.24% vs 48.51%).
>
> ## 5. Unit Test at test time
> The model first generates an initial solution for a given problem description. The initial problem description contains one or more test case examples (example input and expected output).
>
> Then the initial solution is tested against all the test cases provided for the problem. If one test case fails, the model will take the failed test case and the error message to generate refinement.
>
> That is, at the test time, the generation of the initial solution will see some example test inputs and outputs. And the generation of refinement (and explanation) will see the exact failed test cases.
>
> ## 6. Reward Design
> CodeBLEU: We find that only using binary execution feedback as a reward does not train the model properly as the reward is too sparse. That is, a completely wrong solution will get the same reward as an almost correct solution, which hurts the RL training. Although the CodeBLEU reward is weak, we find that having it helps stabilize the training by densifying the reward distribution. This is the main reason we introduce the CodeBLEU score in the reward.
>
> Similarity: The formula for R(e) might look strange, but the main goal is to scale the majority explanation similarity (Figure 3 c in paper) to the range of [-5, 5] so that it is in the same value range as the code rewards (Figure 3 d in paper). Our reward design is based on the statistics of the training data.
>
> ## 7. Generalization
> To avoid overfitting, we use a large batch size (128) to only update the model for a few thousand steps. We test our models’ perplexity on 10000 samples from BigQuery Python code. The CodeLlama-7B pre-trained model’s perplexity is 1.457, the SFT/RL models' are 1.597 and 1.599, just slightly higher.

---

> > ### Comment · Reviewer_osNA · 2024-08-13
> > **Response to authors**
> >
> > Thank you for your response.
> >
> > Can you please edit your response and mention which part of my review does each part in your response refer to?
> >
> > For example, I did not ask for a *comparison* to the teacher models.
> >
> > I also asked:
> > >>2. In the definition of $S_{cb}$ - if we rely on unit tests to verify correctness, why do we need to encourage the refinement to be similar (in terms of CodeBLEU) to **all** the possible correct refinements?
> >
> > The authors' response seems to be answering a different question. My emphasis was on the word "all" - why does it make sense to encourage the refinement to be similar to **all** the possible correct refinements? What if there are multiple ways to solve the same problem?
> >
> > Regarding generalization, the authors mention that the perplexity increases from 1.457 to 1.597, which might be evidence that the model's coding ability is indeed hurt. How can we be convinced that the model does not over-specialize on specific domains?

---

> > > ### Author Response · Authors · 2024-08-13
> > >
> > > Thank you for following up.
> > >
> > > We re-organized the rebuttal so that the response to each question is clear. Due to character limits, we abbreviate the reviewer’s questions, and refer some questions to our first response. Below, we focus primarily on addressing the reviewer’s new questions.
> > >
> > > ## 1. Paper novelty
> > > Reviewer: “The paper is mostly applicative, and includes a mix of known techniques. I feel that the paper is low on conceptual novelty.”
> > >
> > > Response:
> > > Please refer to the “Paper novelty” in our first response.
> > >
> > > ## 2. Position in literature
> > > Reviewer: “Position in literature - Although many papers are cited, I feel that the paper does not position itself well in the literature.”
> > >
> > > Response:
> > > Please refer to the “Position in literature” in our first response.
> > >
> > > ## 3. Self-taught refinement
> > > Reviewer: “This adds a dimension of distillation (from large models to small models), and further makes it difficult to pinpoint the exact source of contribution.”
> > >
> > > Response:
> > > Please refer to “Self-taught refinement” in our first response.
> > > ## 4. Comparison to the teacher models or related work
> > > Reviewer: “The approach is not compared to any related work, or to the teacher models themselves.”
> > >
> > > Response:
> > > The prompting baseline mentioned in our paper refers to the related work [1], and we compare with prompting method in our experiments.
> > >
> > > We provide the comparison with the teacher models CodeLlama-34B and GPT-3.5-Turbo on the self-debugging setup in the global rebuttal pdf. Comparing it with Table 2 and Table 10 in the paper, we see that with CodeLlama-34B as the teacher, CodeLlama 7B SFT/RL achieves close to CodeLlama-34B self-debugging performance, while CodeLlama-13B SFT/RL significantly outperforms the CodeLlama-34B teacher (e.g. in HumanEval+ pass@1 56.24% vs 48.51%).
> > >
> > > ## 5. Unit Test at test time
> > > Reviewer: “At test time, when evaluating the SFT/RL models: does the model see the execution results of its generated test code before refinement?”
> > >
> > > Response: please refer to “Unit Test at test time” in our first response.
> > >
> > > ## 6. Reward Design
> > > Reviewer: “why does it make sense to encourage the refinement to be similar to all the possible correct refinements?”
> > >
> > > Response:
> > > If we only consider one correct solution, there will be correct refinement that uses a different way but get very low CodeBLEU score. And it is also unreasonable to train the model to follow only one correct solution. Our solution does not penalize the model too much when it uses a different way to solve the problem, as long as there exist correct solutions using the similar way.
> > >
> > > Besides, the CodeBLEU score is mainly used to densify the reward distribution, unit test is more important to separate the wrong and correct solutions.
> > >
> > > ## 7. Generalization
> > > Reviewer: “RL training on benchmark data may over-specialize on their specific domain, while degrading the general coding abilities of the LLM.”
> > >
> > > Response:
> > > We have already considered the practical usage of the proposed training, and all of our experiments in the paper use rather comprehensive data, not only the self-debugging data we collected but also the original code generation data provided in the MBPP, APPS, CodeContests training set and Magicoder dataset [2] to avoid over-specialization. Besides, to avoid overfitting, we use a large batch size so the model weights are only updated for about 2000 steps..
> > >
> > > We test our models’ perplexity on general code, e.g., 10000 samples from BigQuery Python code. The CodeLlama-7B pre-trained model’s perplexity is 1.457, the SFT model’s perplexity is 1.597, and the RL model’s perplexity is 1.599. Both are just slightly higher than the pre-trained model. **Code generation is one of the most important code tasks for LLMs and our trained model is much better than the pre-trained model on it.**
> > >
> > > **The SFT/RL models’ perplexity on the pretraining data being higher than the pretrained model doesn’t mean they generalize worse. The SFT (or instruction-tuned) LLMs typically have higher perplexity than the pre-trained foundation model, since they learn to follow human instructions.** We also test the instruction-tuned CodeLlama-7B released by Meta (https://huggingface.co/codellama/CodeLlama-7b-Instruct-hf), which gets a perplexity of 1.682, even higher than ours. We generally don’t see concerns regarding the higher perplexity caused by instruction-tuning, because instruction-tuned models follow users’ instructions better and are more useful in developing AI-assistant.
> > >
> > > Reference:
> > >
> > > [1] Xinyun Chen, Maxwell Lin, Nathanael Schärli, and Denny Zhou. Teaching large language models to self-debug, ICLR 2024.
> > >
> > > [2] Wei, Y., Wang, Z., Liu, J., Ding, Y., & Zhang, L. (2024). Magicoder: Empowering code generation with oss-instruct. In Forty-first International Conference on Machine Learning.

---

> > > > ### Comment · Reviewer_osNA · 2024-08-13
> > > > **Response to authors**
> > > >
> > > > Thank you for your thorough response!
> > > >
> > > > All my concerns are now resolved and I increasedy score to 7. Good luck!

---

### Official Review · Reviewer_uGRw · 2024-07-12

**Soundness:** 4
**Presentation:** 3
**Contribution:** 3
**Rating:** 7
**Confidence:** 5

**Summary:**

This work proposes a novel framework to enhance the self-debugging capabilities of smaller language models that do not benefit much from self-refine or other prompt-based debugging approaches. Sampling incorrect code samples produced by LMs, they pass the execution feedback on these to GPT-3.5/4 and prompt it to explain the reason for the errors and propose code refinements. Accurate refinements are used to fine-tune and create a code-correcting model. This is further enhanced with a PPO based learning from a novel reward assignment mechanism accounting for both explainability and code refinement. Overall, they demonstrate the importance of having explanations for incorrect codes and how RL can be used to enhance the debugging ability of models to show superior performance across benchmark datasets.

**Strengths:**

- Technically solid
- This work addresses a significant issue of little coding improvement from self-refinement prevalent in smaller LMs
- The reward setup incorporating both code refinement and explainability in PPO is novel and shows good gains

**Weaknesses:**

Nothing major, future evaluation on datasets like APPS etc. could be beneficial to understand the impact on harder tasks

**Questions:**

- Was there any reason behind choosing the range of score to be [-5,5]?
- It seems in the bigger models tested (\geq 13B) the refine itself is quite effective, any intuitions on this scale effect?

**Limitations:**

Yes

---

> ### Author Rebuttal · Authors · 2024-08-07
>
> We thank the reviewer for the positive comments.
>
> ## 1. Evaluation of APPS and CodeContests
> This is a good suggestion and we plan to add the results to the final version if accepted.
>
> Below are the results on APPS and CodeContests. We test StarCoder on the full 5000 APPS test samples. However, due to the large amount of test samples in APPS (5000), we only test the CodeLlama-7B/13B on a subset of 200 samples. We plan to keep running to complete the evaluation and add it to the paper.
>
> | Approaches | | APPS (5000) | | CodeContests (165) | |
> |-|-|-|-|-|-|
> | | | Pass@1 | Pass@10 | Pass@1 | Pass@10 |
> | | Init. | 2.57 | 8.59 | 0.58 | **3.88** |
> | Prompt | Refine | 2.84 | 9.10 | 0.65 | 4.23 |
> | | Expl. + Ref. | 2.95 | 9.53 | 0.78 | 4.84 |
> |||||||
> | | Init. | 3.80 | 11.52 | **0.62** | 3.67 |
> | SFT | Refine | 6.89 | 17.01 | 1.16 | 5.25 |
> | | Expl. + Ref. | 6.86 | 17.18 | 1.48 | 5.94 |
> |||||||
> | | Init. | **4.50** | **13.62**  | 0.44 | 2.19 |
> | RL | Refine | **7.81** | **19.28** | **1.80**  | **6.04** |
> | | Expl. + Ref. | **8.10** | **19.75** | **1.80** | **6.37** |
>
> On APPS and CodeContests, the prompting baseline also only refines very few solutions and the improvements brought by refinement are marginal. However, with our SFT and RL-trained models, we see significantly stronger self-refinement ability. The final Pass@k is also about doubled from the prompting baseline.
>
> ## 2. Reward Range
> Our RL algorithm is based on the PPO algorithm. According to the details in Appendix A3, the rewards of each token are either the code reward, explanation reward, or the KL divergence. From our calculation of the KL divergence distribution on a subset of the training data, the trajectory tokens’ KL divergences have a minimum value of -5.22 and a maximum value of 5.37. Thus, we scale our code and explanation rewards to the range of [-5, 5], which is similar to the KL divergence value range.
>
> Actually, we think as long as the code/explanation rewards are not too large or too small to cause gradient explosion, the setup should be reasonable. For example, [-1, 1], [0, 1], are also common choices of reward range.
>
> ## 3. Scale Effect
> We observed a similar scaling effect in Table 4 in Section 4.1.2 where we show the successful refinement rate of different models across prompting, SFT, and RL. We see that from CodeLlama 7B to CodeLlama 13B, the refinement rate improves by around 3% (absolute rate) across the board for all approaches. We think that larger models have better capabilities in general, and the trend applies to self-debugging capability as well. Our proposed training seems to help push the self-debugging performance closer to the model’s limit, and in general, a larger model should have a higher upper bound.

---

### Official Review · Reviewer_2CSS · 2024-07-12

**Soundness:** 3
**Presentation:** 3
**Contribution:** 3
**Rating:** 5
**Confidence:** 4

**Summary:**

The authors teach language models to better self-debug and explain code. Particularly, they utilize code explanations in the repair process where explanations are generated before refining the programs. This is accomplished via training the models with SFT and RL on data curated from different sources and model generations with test case based rejection sampling.

**Strengths:**

The paper is nicely written with enough details about experiments. Improving code repair capabilities of LLMs is an important problem and the authors propose a rejection-sampling-based technique to automatically curate repair trajectories from LLMs. The associated ablations are useful and convey useful insights.

**Weaknesses:**

Weak results
* Benefit of explain + refine over just refine.
The proposed explain + refine approach, associated loss on explanations does not seem to be improving performance. Infact, Expl.+Refine rows sometimes do worse than the only Refine rows. These results highlight a lack of benefit from adopting this approach. While I understand that having associated explanations is appealing and perhaps might even provide models with more inference-time-compute before performing refinements, I think the current approach or experiments does not convey that. The associated human evaluations (Table 7) also point to similar findings.

* Optimizing single-turn vs multi-turn performance.
Finally, a lot of the instruction-tuned variants of the models used in this work claim better single-turn HumanEval and MBPP performance. I wonder how the findings would change if they started from a strong instruction-tuned model and improved its repair performance. For instance, the OpenCodeInterpreter-CL-7B model shows performance improvements of 3 points from execution feedback on HumanEval (from 72 to 75) due to strong instruction tuning data pushing pass@1 to 72. Perhaps to a larger point, the effect of repair depends on the choice of the underlying model and broader data mixture which this paper does not study.

* Choice of evaluation datasets.
Since the authors use competition programs (like APPS or CodeContests) in their study, perhaps it is fair to also evaluate the models on competition programming datasets.

* Missing related works.
[1] is also pertinent to LLM reasoning. [2] and [3] also train and release open LLMs on repair trajectories. [3] in fact uses a similar explanation format.

[1] Reflexion: Language Agents with Verbal Reinforcement Learning

[2] OpenCodeInterpreter: Integrating Code Generation with Execution and Refinement

[3]  Advancing LLM Reasoning Generalists with Preference Trees

**Questions:**

* Table 1 further details. I anticipate that many problems in the training sets remain unsolved post-refinement, particularly for APPS and CodeContests which are more challenging benchmarks. Can the authors list details about problems beyond number of solutions.

**Limitations:**

Limitations are discussed adequately (besides some of the above-mentioned weaknesses)

---

> ### Author Rebuttal · Authors · 2024-08-07
>
> We thank the reviewer for insightful suggestions and questions.
>
> ## 1. Explanation improvement
> The reviewer might have some misunderstanding of the explanation evaluation in Table 7. In Table 7, we performed both human evaluation and LLM Judge-based evaluation to evaluate the explanation quality before and after training. It clearly shows that the SFT/RL training significantly improves the explanation quality, by generating more correct and helpful bug explanations. Appendix A6 Table 13 lists the rubrics.
>
> We also provide some case studies on human evaluation in Appendix A6 of the paper to show qualitatively how the finetuned model generates better explanations.
>
> ## 2.Comparison with Instruction-tuning
> In Appendix A.4.1 of the paper, we have presented the comparison of SFT on strong code instruction data with SFT on the full data we collected. The code instruction data we used includes the Magicoder dataset [ref], similar to the OpenCodeInterpreter-CL-7B the reviewer mentioned. The results in Appendix A.4.1 show that even when training with strong code instruction data, the model does not obtain self-debug capability out-of-the-box and it only refines up to 3% of its wrong solutions. And this is one of the main motivations of the proposed approach.
>
> ## 3. Evaluation of APPS and CodeContests
> This is a good suggestion and we plan to add the results to the final version if accepted.
>
> Below are the StarCoder-15B’s results on APPS and CodeContests. Results of the other two backbones can be found in our attached author response PDF file (Tables 4 and 5). Due to the large amount of test samples in APPS (5000), we only test the other two backbones on a subset of 200 samples. We plan to complete the evaluation and add it to the paper.
>
> | Approaches | | APPS (5000) | | CodeContests (165) | |
> |-|-|-|-|-|-|
> | | | Pass@1 | Pass@10 | Pass@1 | Pass@10 |
> | | Init. | 2.57 | 8.59 | 0.58 | **3.88** |
> | Prompt | Refine | 2.84 | 9.10 | 0.65 | 4.23 |
> | | Expl. + Ref. | 2.95 | 9.53 | 0.78 | 4.84 |
> |||||||
> | | Init. | 3.80 | 11.52 | **0.62** | 3.67 |
> | SFT | Refine | 6.89 | 17.01 | 1.16 | 5.25 |
> | | Expl. + Ref. | 6.86 | 17.18 | 1.48 | 5.94 |
> |||||||
> | | Init. | **4.50** | **13.62**  | 0.44 | 2.19 |
> | RL | Refine | **7.81** | **19.28** | **1.80**  | **6.04** |
> | | Expl. + Ref. | **8.10** | **19.75** | **1.80** | **6.37** |
>
> On APPS and CodeContests, the prompting baseline also only refines very few solutions, and the improvements brought by refinement are marginal. However, with our SFT and RL-trained models, we see significantly stronger self-refinement ability. The final Pass@k is also about doubled from the prompting baseline.
>
> ## 4. Related works
> We thank the reviewer for pointing out missing related works and providing some discussions here.
>
> In the Reflexion paper, an LLM verbally reflects on task feedback signals and generates improvements. It is similar to the prompting method.
>
> OpenCodeInterpreter constructs a multi-turn interaction dataset that integrates execution and human feedback for code refinement using GPT-4, and shows that models on such dataset achieve good refinement performance.
>
> In EURUS paper ([3] mentioned by the reviewer), it curates an UltraInteract dataset as an alignment dataset for complex reasoning tasks using GPT-3.5-Turbo and GPT-4, and includes multi-turn interaction trajectories with the environment and the critique.
>
> Both of the last two papers obtain multi-turn feedback interaction dataset using close models GPT-3.5-Turbo and GPT-4, and perform finetuning. Different from them, our paper shows that even without strong LLMs like GPT-3.5-Turbo and GPT-4, we can generate synthetic data for self-debugging using much smaller open pretrained/instruct models or the same LLM itself from the additional results from this rebuttal, and they are quite effective. We also proposed a novel RL training with explanation and execution reward in our paper.
> ## 5. Details on training data
> Here is a summary of the APPS dataset and the number of problems that has at least one correct solution (either correct initial solution or correct refinement from GPT-3.5-Turbo) based on difficulty levels:
>
> | | Introductory | Interview | Competition |
> |-|-|-|-|
> | Count | 2410 / 2639 | 493 / 2000 | 306 / 361 |
>
>
> For the CodeContests dataset, the data is from multiple sources and difficulty levels are not comparable. We give the sample distribution from CodeChef with difficulty levels 1-4:
>
> | | EASY (1)  | MEDIUM (2) | HARD (3) | HARDER (4) |
> |-|-|-|-|-|
> | Count | 19 / 86 | 65 / 330 | 18 / 90 | 1 / 6 |
>
> Problems without any correct initial solutions or refinements are discarded from the training data.

---

> ### Comment · Reviewer_2CSS · 2024-08-09
>
> Thanks for the response.
>
> > The reviewer might have some misunderstanding of the explanation evaluation ... SFT/RL training significantly improves the explanation quality
>
> From what I can see, the absolute ratings from developers is below 3 on average on a 1-5 scale. While the performance improvement from untrained model baseline is considerable, the absolute scores are still not great.
>
> I suspect a reason why absolute numbers are low is that not all explanations lead to refinement and the average rating might be higher in the second scenario.
>
> >  The results in Appendix A.4.1 show that even when training with strong code instruction data, the model does not obtain self-debug capability out-of-the-box and it only refines up to 3% of its wrong solutions
>
> Perhaps I was not clear enough -- this work depicts considerable mutliturn improvements from RL training (48% to 57% pass@1 on HumanEval for CodeLLama-7B). My concern is that if the authors start their RL training with a stronger model (say OpenCodeInterpreter-CL-7B ) which already has 70+ pass@1 on HumanEval will the RL training be as effective? For example, OpenCodeInterpreter-CL which trains models with multi-turn SFT dataset only achieves 3% improvements from 72 to 75. This makes it challenging to interpret the performance improvements achieved in this paper.

---

> > ### Author Response · Authors · 2024-08-10
> >
> > Thank you for following up.
> >
> > ## Human Rating
> > Below is the breakdown of the number of explanations having the score in each range. Although the overall average score is below 3, SFT and RL can generate 24 and 27 explanations with scores higher than 3 (50 samples in total).
> >
> > | Score | Prompt | SFT | RL | GPT-3.5-Turbo |
> > |-|-|-|-|-|
> > | 4.5 <= score | 1 | 3 | 5 | 13 |
> > | 4 <= score | 1 | 10 | 11 | 21 |
> > | 3.5 <= score | 3 | 16 | 17 | 26 |
> > | 3 <= score | 7 | 24 | 27 | 35|
> >
> > If we look at poor explanations, the RL model generates a similar number of explanations with “score <= 1.5” compared to GPT-3.5.
> >
> > | Score | Prompt | SFT | RL | GPT-3.5-Turbo |
> > |-|-|-|-|-|
> > | score == 1 | 19 | 6 | 4 | 6 |
> > | score <= 1.5 | 33 | 14 | 10 | 9 |
> > | score <= 2 | 39 | 21 | 21 | 14 |
> > | score <= 2.5 | 43 | 26 | 23 | 15 |
> >
> > We also find that human annotators tend to be more harsh and give lower scores than GPT-4. Figures 12 and 13 include examples of explanations with “score 4”, which we think the model explains pretty well.
> >
> >
> > ## SFT/RL Improvement
> > There seems to be some misalignment from the number you mentioned.
> >
> > If we look at Table 2, HumanEval Pass@1 using CodeLlama-7B. With code explanation and refinement (Expl. + Refine.), prompting’s Pass@1 is 40.13%, SFT’s Pass@1 is 52.98%, RL’s Pass@1 is 55.84%. So SFT increases it by about 13%, and RL (on top of SFT) further increases it by about 3%.
> >
> > And also by looking at Table 9, with only single-turn data (MBPP, APPS, CodeContests, MagiCoder’s data), the Pass@1 on HumanEval is 43.88%. SFT (with multi-turn refinement data) can still get 9% improvement.
> >
> > For OpenCodeInterpreter, it has been trained on high-quality data collected using GPT-4, so the base Pass@1 is already high enough (72%), which may nearly reach the model’s limit and leave less space for mult-turn SFT refinement to improve. That could be why multi-turn SFT only improves 3%. **We think this is not contradictory with our results.**
> >
> > Our focus is on how to train LLM to self-debug starting from standard code generation training data (MBPP, APPS, CodeContests). Thus we only use GPT-3.5 as the teacher to collect multi-turn data. We also test by using CodeLlama-34B as the teacher (Table 5), and even using CodeLlama-7B itself to self-bootstrap multi-turn data (mentioned in global response). Our method can work without relying on GPTs.
> >
> > This work is not competing on data quality, so the final results are not supposed to outperform OpenCodeInterpreter. But we would like to add OpenCodeInterpreter as related work and discuss it in our paper.

---

> > > ### Comment · Reviewer_2CSS · 2024-08-10
> > >
> > > Thank you for the quick response.
> > >
> > > > If we look at Table 2, HumanEval Pass@1 using CodeLlama-7B. With code explanation and refinement (Expl. + Refine.), prompting’s Pass@1 is 40.13%, SFT’s Pass@1 is 52.98%, RL’s Pass@1 is 55.84%. So SFT increases it by about 13%, and RL (on top of SFT) further increases it by about 3%.
> > > >
> > > > And also by looking at Table 9, with only single-turn data (MBPP, APPS, CodeContests, MagiCoder’s data), the Pass@1 on HumanEval is 43.88%. SFT (with multi-turn refinement data) can still get 9% improvement.
> > >
> > > Apologies for the misquoted numbers and thanks for the clarification. I will update my rating.

---

### Official Review · Reviewer_Dkve · 2024-07-18

**Soundness:** 3
**Presentation:** 4
**Contribution:** 3
**Rating:** 8
**Confidence:** 4

**Summary:**

This paper proposed a pipeline to obtain code explanation and refinement data from a stronger model (mainly GPT-3.5 with CodeLLaMA as ablation) to train weaker models (i.e., StarCoder-15B, CodeLLaMA-7B/13B) using SFT and RL methods. More specifically, it uses the weaker models to sample the incorrect solutions and use the stronger models to give code refinement and explanations. It also designed reinforcement learning rewards specifically for the code refinement and explanation tasks. Experiments are conducted on MBPP and HumanEval, as well as their EvalPlus versions, and results show that the both SFT and RL yields improvements over the baselines, while the improvements from RL are relatively marginal compared to those of SFT.

**Strengths:**

S1. The tasks of code refinement and explanation are increasingly important due to the popularity of using language model agents for coding tasks. And this work shows a way to reliably improve the performance of LLMs in these two tasks, which may be useful for broader domains such as code editing and reasoning;
S2. The RL reward design and ablations could be useful for further research on using RL for code explanations and refinements;
S3. The paper is well-written, with clear motivations, details of methodology and comprehensive experiments.

**Weaknesses:**

W1. Something slightly disappointing is that this work chooses to generate training data **from a stronger model**, which classifies it into the category of distillation, while the entire pipeline could have been done with the same LLM to explore the potential of self-improvement;
W2. While the paper focus a lot on the design of the RL method (which seems to be a big part of the contribution from my understanding), the actual improvements yield by the RL methods are quite marginal. However, I do note that at least the performance does not decrease for the most cases;
W3. There are some baselines experiments and ablations that can be added to make the evaluation part stronger (see questions below).

**Questions:**

Q1. According to Olausson et al., 2023 on self-repair, after $k$ rounds of self-refinement after the initial code generation, the success rate is often less than simply pass@$k+1$. And also note that the sampling can be done in parallel while the iterative refinement can only be done sequentially. Have you compared the refinement results with pass@$k+1$ to see if self-refinement yields any benefit before / after the SFT/RL training?
Q2. From Fig. 3(a), it seems to me that the CodeBLEU score is not a good metric as it can barely separate the correct and wrong outputs distribution-wise, is there any reason for CodeBLEU to still be factorized into the reward function despite this?
Q3. Can you comment on the reliability of using the RoBERTa embedding for measuring the similarity of the explanations? Are there better ways to do this?
Q4. From Tab. 2, it seems that after SFT, the "Init." performance also significantly improved, does this means that **only** training on code refinement and explanation can also improve code generation?
Q5. (Pertain to W3) Have you tried to use the same model as the LLM to create the training data?
Q6. The training data are created on top of APPS and CodeContest as well, but why are those two datasets not used in evaluation?

**References**
Olausson, Theo X., et al. "Is Self-Repair a Silver Bullet for Code Generation?." The Twelfth International Conference on Learning Representations. 2023.

---

> ### Author Rebuttal · Authors · 2024-08-06
>
> We thank the reviewer for insightful suggestions and questions.
>
> ## 1. Data collected from the same model itself
> We provide the experiment results using the synthetic data generation from the model itself for CodeLlama 7B.
>
> We highlight some results here. Below is the CodeLlama-7B SFT/RL using the data collected from itself, evaluated on MBPP+ and HumanEval+.
>
> | Approaches | | MBPP+ | | HumanEval+ | |
> |-|-|-|-|-|-|
> | | | Pass@1 | Pass@10 | Pass@1 | Pass@10 |
> | | Init. | 37.18 | 61.23 | 27.40 | 60.81 |
> | Prompt | Refine | 42.97 | 66.89 | 31.84 | 65.08 |
> | | Expl. + Ref. | 42.46 | 67.41 | 32.49 | 66.58 |
> |||||||
> | | Init. | 41.78 | 61.77 | 33.25 | 61.50 |
> | SFT | Refine | 46.26| **66.39** | 40.15 | 67.15 |
> | | Expl. + Ref. | 45.94 | 65.77 | 39.10 | 67.33 |
> |||||||
> | | Init. | 41.61 | 61.29 | 33.66 | 62.17 |
> | RL | Refine | **46.28** | 65.86 | **41.54** | 68.14 |
> | | Expl. + Ref. | 46.10 | 65.99 | 40.79 | **68.50** |
>
> The full results are in our attached author response PDF file, Tables 1 and 2. Results show that self-taught SFT and RL also achieve big improvements. CodeLlama-7B SFT/RL models achieve up to 5% improvement in self-debugging, compared with the baseline prompting method, and the model trained with code generation data only. Though comparing the experiments using data from CodeLlama 34B and GPT-3.5-Turbo, the improvement is less.
>
> ## 2. Pass@K+1 versus refinement Pass@K
> This is a very interesting point. We evaluate the Pass@2 of the initial solution and Pass@1 after one round of refinement.
>
> Below is the result when the models are not trained (the prompting baseline). We do observe that Pass@2 is better than Pass@1 after refinement. **This shows that using a prompting approach to self-debug is not effective**.
>
> | Before Training | | MBPP+ | HumanEval+ |
> |-|-|-|-|
> | StarCoder-15B | Pass@2 | **45.20** | **36.38** |
> | | Expl. + Ref. Pass@1 | 39.27 | 30.09 |
> | CodeLlama-7B | Pass@2 | **46.91** | **38.17** |
> | | Expl. + Ref. Pass@1 | 42.46 | 32.49 |
> | CodeLlama-13B | Pass@2 | **48.08** | **41.52** |
> | | Expl. + Ref. Pass@1 | 45.77 | 38.36 |
>
> However, after the models are trained using our pipeline. The refinement Pass@1 is clearly higher than the Pass@2 of initial solutions. This shows that the model’s self-debugging performance is bad without training. The prompting approaches proposed by existing works such as (https://arxiv.org/pdf/2306.09896 and https://arxiv.org/abs/2304.05128) are not as effective on open-sourced LLMs. **This experiment further supports our motivation to train LLMs to self-debug and proves the effectiveness of our approach.**
>
> | After SFT | | MBPP+ | HumanEval+ |
> |-|-|-|-|
> | StarCoder-15B | Pass@2 | 51.19 | 39.29 |
> | | Expl. + Ref. Pass@1 | **53.83** | **43.54** |
> | CodeLlama-7B | Pass@2 | 50.93 | 40.95 |
> | | Expl. + Ref. Pass@1 | **51.55** | **47.62** |
> | CodeLlama-13B | Pass@2 | 50.93 | 44.78 |
> | | Expl. + Ref. Pass@1 | **54.59** | **51.32** |
>
> ## 3. CodeBLEU in RL reward
> We find that only using binary execution feedback as a reward does not train the model properly as the reward is too sparse. That is, a completely wrong solution will get the same reward as an almost correct solution, which hurts the RL training. Although the CodeBLEU reward is weak, we find that having it helps stabilize the training by densifying the reward distribution. This is the main reason we introduce the CodeBLEU score in the reward.
>
> ## 4. Roberta embedding for text similarity
> Judging the correctness of code explanation is non-trivial. We use the Roberta model (https://huggingface.co/sentence-transformers/all-roberta-large-v1) that has been massively fine-tuned for text similarity using 1B sentence pairs.
>
> We try to analyze the reliability of this approach as shown in Figure 3(c) in the paper. The similarity can separate the explanations that lead to correct and wrong solutions most of the time.
>
> A potential alternative approach could be using powerful LLMs such as GPT-4 to rate the code explanation, however, this is not scalable enough the handle our RL training data.
>
> ## 5. Code generation improvement
> It should be noted that the SFT training includes code generation data (provided in the original MBPP/APPS/CodeContests) + self-debug data. It could be the code generation data improves the initial solution generation. We will make this clearer in our experiment setup.
>
> We compare our approach with fine-tuning using purely code-generation data. The results are in Appendix A4.1 Table 9 in the paper.
>
> ## 6. Evaluation of APPS and CodeContests
> This is a good suggestion and we plan to add the results to the final version if accepted.
>
> Below are the StarCoder-15B’s results on APPS and CodeContests. Results of the other two backbones can be found in our attached author response PDF file (Tables 4 and 5). Due to the large amount of test samples in APPS (5000), we only test the other two backbones on a subset of 200 samples. We plan to complete the evaluation and add it to the paper.
>
> | Approaches | | APPS (5000) | | CodeContests (165) | |
> |-|-|-|-|-|-|
> | | | Pass@1 | Pass@10 | Pass@1 | Pass@10 |
> | | Init. | 2.57 | 8.59 | 0.58 | **3.88** |
> | Prompt | Refine | 2.84 | 9.10 | 0.65 | 4.23 |
> | | Expl. + Ref. | 2.95 | 9.53 | 0.78 | 4.84 |
> |||||||
> | | Init. | 3.80 | 11.52 | **0.62** | 3.67 |
> | SFT | Refine | 6.89 | 17.01 | 1.16 | 5.25 |
> | | Expl. + Ref. | 6.86 | 17.18 | 1.48 | 5.94 |
> |||||||
> | | Init. | **4.50** | **13.62**  | 0.44 | 2.19 |
> | RL | Refine | **7.81** | **19.28** | **1.80**  | **6.04** |
> | | Expl. + Ref. | **8.10** | **19.75** | **1.80** | **6.37** |
>
> On APPS and CodeContests, the prompting baseline also only refines very few solutions, and the improvements brought by refinement are marginal. However, with our SFT and RL-trained models, we see significantly stronger self-refinement ability. The final Pass@k is also about doubled from the prompting baseline.

---

> > ### Comment · Reviewer_Dkve · 2024-08-13
> > **Thanks for the response**
> >
> > I'd like to thank the authors for the detailed response, especially the additional experiment results.
> >
> > I find the additional results interesting and promising, and I think the argument on using CodeBLEU stabilizes training makes a lot of sense.
> >
> > I think adding the results / discussions from at least 1+2+3 from above would make the paper more interesting and stronger, so I hope the authors would add them in the next version of the paper.
> >
> > I have improved my score accordingly, good luck!

---

### Official Review · Reviewer_jfHT · 2024-07-24

**Soundness:** 3
**Presentation:** 3
**Contribution:** 3
**Rating:** 6
**Confidence:** 4

**Summary:**

Authors propose a framework to perform SFT and RL training to achieve superior performance in generating code with open code LMs on the self-debugging task. They leverage the test suite present in benchmarks like APPS, CodeContests to obtain execution feedback for model refinements on CodeLM generated code. The resulting dataset is then used for SFT training, followed by RL training that can also leverage failed trajectories besides the successful ones (where the model succeeded in generating a code that passes all unit tests). Authors show promising gains for open weight models in the model's capability to self-debug after training with their proposed methodology.

**Strengths:**

- Authors present a strong motivation for this work (Lines 43-52) on achieving strong code generation performance with open weight models.
- This work makes a strong contribution in the form of a framework to construct data for SFT and RL training of a model that can perform self-debugging after explaining faulty code. Authors propose a clever way of leveraging execution feedback in constructing their datasets
- The reward construction based on environment feedback is a particularly important contribution that significantly adds to the novelty of this work. To my knowledge prior work hasn't utilised environment feedback in this manner.
- Convincing results that confirm the utility of training open codeLMs on the task of self-debugging.

**Weaknesses:**

- While the experiments and analysis of the results and datasets are fairly exhaustive, I believe the choice of RL algorithms should be justified by considering or eliminating alternatives like preference optimisation using DPO or KTO. I'd suggest at least adding a discussion on the pros and cons of preference learning compared to the RL setup that the authors advocate in this work.
- Some missing baselines: teacher model (GPT-4/3.5/CodeLlama35B) performance is missing in Tables 2 and 5. Authors do not discuss the persisting gap in performance in any between the models used in creating the datasets and the performance their approach attains on the benchmarks used.
- A very relevant related work (Teaching LLMs to Self-Debug https://arxiv.org/abs/2304.05128) mentions gains in sample efficiency as one of the major benefits of performing self-debugging/refinement. I could not find a discussion or results on this aspect for the fine-tuned models presented in this paper.

**Questions:**

- APPS and CodeContests are used to train the CodeLMs, but I could not find evaluation on the APPS-test or CodeContests test set. Could you explain this choice?
- Can you provide details on the number of GPU hours required in the experiments?
- Have the authors considered training a separate model using the SFT and RL techniques to solely solve code refinement for code generated from the base model?
- What do the authors think about the generality of this method to improve on attributes beyond correctness in refining code? e.g. readability, performance and secureness of generated code.
- I'm curious how the proposed approach would compare against a simple baseline where the model is SFT-trained on the final refinement collected in your training set. The current setup involves using problem description $x$ and ground truth code $y$, test suite $T$ to generate a synthetic code solution $y'_l$ that fails and is explained by $x_l$ followed by a successful refinement $y'_w$. Your approach then trains the model to generate $x_l, y'_w$ given $x, y, y'_l$. A simple baseline to compare against could involve training the model to generate $y'_w$ given $x$. This would confirm the value in framing code generation as a explanation + refinement task.

**Limitations:**

Briefly discussed by the authors in Section 5. Could be expanded to include a discussion on other aspects of code refinement not covered in the paper, and acknowledging gaps if any in performance of open models trained with this method and closed models.

---

> ### Author Rebuttal · Authors · 2024-08-06
>
> We thank the reviewer for insightful suggestions and questions.
>
> ## 1. APPS and CodeContests Evaluation
> This is a good suggestion and we plan to add the results to the final version if accepted.
>
> Below are the StarCoder-15B’s results on APPS and CodeContests. Results of the other two backbones can be found in our attached author response PDF file (Tables 4 and 5).
>
> | Approaches | | APPS (5000) | | CodeContests (165) | |
> |-|-|-|-|-|-|
> | | | Pass@1 | Pass@10 | Pass@1 | Pass@10 |
> | | Init. | 2.57 | 8.59 | 0.58 | **3.88** |
> | Prompt | Refine | 2.84 | 9.10 | 0.65 | 4.23 |
> | | Expl. + Ref. | 2.95 | 9.53 | 0.78 | 4.84 |
> |||||||
> | | Init. | 3.80 | 11.52 | **0.62** | 3.67 |
> | SFT | Refine | 6.89 | 17.01 | 1.16 | 5.25 |
> | | Expl. + Ref. | 6.86 | 17.18 | 1.48 | 5.94 |
> |||||||
> | | Init. | **4.50** | **13.62**  | 0.44 | 2.19 |
> | RL | Refine | **7.81** | **19.28** | **1.80**  | **6.04** |
> | | Expl. + Ref. | **8.10** | **19.75** | **1.80** | **6.37** |
>
> On APPS and CodeContests, the prompting baseline also only refines very few solutions, and the improvements brought by refinement are marginal. However, with our SFT and RL-trained models, we see significantly stronger self-refinement ability. The final Pass@k is also about doubled from the prompting baseline.
> ## 2. Preference learning (DPO/KTO) vs PPO
> Preference learning like DPO or KTO has the advantage of its simplicity without the need for a reward function or reward model. In the setup of self-debugging with execution feedback in this paper, we could construct preference data in such a way: the fix that passes unit tests is preferred over the one that fails unit tests. Such preference data could also work, but might seem less direct as execution feedback/reward is easily obtained from the execution engine, unlike the human preference setup. One advantage of our RL training is that we can assign different rewards to different parts of the sequence, i.e. the explanation reward on the explanation part and the execution reward on the generated fix part.
>
> In this work, our focus is exploring how to train LLMs for self-debugging and bug explanation with the lack of training data. Our proposed data collection pipeline and SFT, RL (PPO-based) training do outperform existing prompting approaches significantly. Exploring alternative RL algorithms could be interesting future work in this domain.
>
> ## 3. Teacher models performance
> We provide the comparison with the teacher models in our attached author response PDF file.
>
> Comparing it with Table 2 in the paper, with GPT-3.5 as the teacher, CodeLlama-13B SFT/RL cannot outperform GPT-3.5, which could be because GPT-3.5 is a much stronger teacher model.
>
> Comparing it with Table 10 in the paper, with CodeLlama-34B as the teacher, CodeLlama-13B SFT/RL **sometimes outperforms the CodeLlama-34B teacher** (e.g. in HumanEval+ pass@1 56.24% vs 48.51%, and in MBPP+ pass@1 56.60% vs 53.19%).
>
> ## 4. Comparison with “Teaching LLMs to Self-Debug”
> **The method in “Teaching LLMs to Self-Debug” is the prompting baseline that this paper refers to in Tables 2, 4, and 7.** We directly compare the results of our proposed method with this baseline in our experiments, and the results show that such a prompting approach cannot work as well on open-sourced LLMs.
>
> The most notable difference is that “Teaching LLMs to Self-Debug” investigates the self-debugging of commercial LLMs by prompting, while our paper investigates how to improve open-source LLMs’ self-debugging capability via training.
>
> Another difference is that our paper tries to **generate the explanation of the bug to help humans and LLMs better understand the reasoning**, while the paper “Teaching LLMs to Self-Debug” tries to generate an explanation for the code instead of reasoning the bug.
>
> ## 5. GPU hours
> The GPU hours for training:
>
> | Models | SFT | RL |
> |-|-|-|
> | StarCoder-15B | 320h | 192h |
> | CodeLlama-7B | 80h | 96h |
> | CodeLlama-13B | 280h | 178h |
>
> Experiments are conducted on 8 NVIDIA A100 GPUs, each with 40GB of memory
>
> ## 6. Generate 𝑦𝑤′ given 𝑥
> If we understand correctly, the suggestion is generating the correct refinement (𝑦𝑤′) given only the problem description (𝑥). We think this is essentially fine-tuning with code generation data.
>
> We compared our approach (framing code generation as an explanation + refinement) with fine-tuning with code generation data only, and the results are shown in Appendix Table 9.
>
> We find fine-tuning for code generation improves the initial solution, achieving comparable or sometimes higher Pass@1 than our approach. **But the model’s self-debugging ability is not improved**, and the model still cannot benefit from self-debugging. The model trained for code generation cannot successfully self-debug very often and eventually is surpassed by our approach dramatically. We hope this confirms the benefit of “framing code generation as an explanation + refinement task” over simply training a better code generation model.
>
> ## 7. Separate code generation and refinement models
> Training a separate dedicated debugging code refinement model is definitely an option. The main motivation of the paper is to improve LLM’s self-debugging capability as one part of the capabilities of LLMs, and not be over-specialized so that the training recipe can directly be incorporated in practice.
>
> ## 8. Generalize to other aspects
> We think that the method can be generalized to improve other aspects of code such as readability and secureness. The key lies in how to properly design the reward with respect to each aspect. The readability and secureness are most likely to be determined by a set of static analysis rules. Combining all aspects together to obtain a final reward is necessary and used to provide feedback to the training.

---

### Author Rebuttal · Authors · 2024-08-07

# Global Response

## 1. Contribution and Novelty
While there are more and more works on self-debugging, most of the existing works focus on how to prompt existing LLMs to do self-debugging. Few works investigate the self-debugging capability of LLMs and how to improve it at the time of submission. Our paper primarily focuses on how to improve the self-debugging capabilities of LLMs via **training**.

We propose a systematic way from synthetic data generation to SFT and RL training with novel explanation and execution rewards. This is the main contribution and novelty of our paper. RL training is important as it helps LLMs to learn from both success and failure generations, and the separation of explanation reward and execution reward helps LLMs learn differently about the explanations and fixes. We also show that LLMs without the proposed training have poor self-debugging capability even if it is trained with strong code-instruction data, and the proposed method is important in improving this capability.

While there seems to be a concurrent work from ICML 2024 on the same topic, there are significant differences and novelties in our paper, for example, the proposed RL training with explanation and execution reward, how we synthesize explanations to explain bug, how we conducted much larger scale experiments such as APPS and CodeContests, and the ablation studies we performed to showcase the importance and effectiveness of our method.

## 2. Self-taught Synthetic Data Generation
Many reviewers are interested in this. The synthetic data generation can also be done on the same model forming a self-taught manner, instead of “distillation” from stronger models. We provide additional experiment results of this kind in the attached PDF for CodeLlama-7B, where the synthetic data is generated from the model itself and it is used to fine-tune the same model.

Experiments show that self-taught synthetic data generation and training also achieve significant improvements. CodeLlama-7B SFT model achieves up to 5% improvement in self-debugging using data generated from CodeLlama-7B, compared with the baseline prompting method and code-instruction-only model. Though comparing the experiments using data from CodeLlama-34B and GPT-3.5-Turbo, the improvement is slightly less.

## 3. Evaluation on APPS and CodeContests
Also many reviewers are interested in this. We provide evaluation results on the test set of APPS and CodeContests with StarCoder-15B below and the other two backbones in the attached PDF file (Tables 4 and 5). On APPS and CodeContests, the prompting baseline also only refines very few solutions, and the improvements brought by refinement are marginal (on APPS 2.57%->2.84%; on CodeContests 0.58%->0.78%). However, with our SFT and RL-trained models, we see significantly stronger self-refinement ability (on APPS 4.50%->8.10%; on CodeContests 0.44%->1.80%), about doubled from the prompting baseline.


| Approaches | | APPS (5000) | | CodeContests (165) | |
|-|-|-|-|-|-|
| | | Pass@1 | Pass@10 | Pass@1 | Pass@10 |
| | Init. | 2.57 | 8.59 | 0.58 | **3.88** |
| Prompt | Refine | 2.84 | 9.10 | 0.65 | 4.23 |
| | Expl. + Ref. | 2.95 | 9.53 | 0.78 | 4.84 |
|||||||
| | Init. | 3.80 | 11.52 | **0.62** | 3.67 |
| SFT | Refine | 6.89 | 17.01 | 1.16 | 5.25 |
| | Expl. + Ref. | 6.86 | 17.18 | 1.48 | 5.94 |
|||||||
| | Init. | **4.50** | **13.62**  | 0.44 | 2.19 |
| RL | Refine | **7.81** | **19.28** | **1.80**  | **6.04** |
| | Expl. + Ref. | **8.10** | **19.75** | **1.80** | **6.37** |

## 4. Comparison with teacher model
We provide the comparison with the teacher models in our attached author response PDF file.

Comparing it with Table 2 in the paper, with GPT-3.5 as the teacher, CodeLlama-13B SFT/RL cannot outperform GPT-3.5, which could be because GPT-3.5 is a much stronger teacher model than our backbones.

Comparing it with Table 10 in the paper, with CodeLlama-34B as the teacher, CodeLlama-13B SFT/RL **sometimes outperforms the CodeLlama-34B teacher** (e.g. in HumanEval+ pass@1 56.24% vs 48.51%, and in MBPP+ pass@1 56.60% vs 53.19%). This is non-trivial and surprising that our approach enables the smaller LLMs to outperform the teacher models sometimes.

## 5. Pass@K+1 versus refinement Pass@K
The reviewer mention a very interesting point: generation followed by one round of refinement may not be better than simply regenerating one more solution. That is, is Pass@K of generation and refinement better than Pass@K+1 of generation only?

We evaluate the Pass@2 of the initial solution and Pass@1 after one round of refinement. Below is the result when the models are not trained (the prompting baseline). We do observe that Pass@2 is better than Pass@1 after refinement. **This shows that using a prompting approach to self-debug is not effective**.

| Before Training | | MBPP+ | HumanEval+ |
|-|-|-|-|
| StarCoder-15B | Pass@2 | **45.20** | **36.38** |
| | Expl. + Ref. Pass@1 | 39.27 | 30.09 |
| CodeLlama-7B | Pass@2 | **46.91** | **38.17** |
| | Expl. + Ref. Pass@1 | 42.46 | 32.49 |
| CodeLlama-13B | Pass@2 | **48.08** | **41.52** |
| | Expl. + Ref. Pass@1 | 45.77 | 38.36 |

However, after the models are trained using our pipeline. The refinement Pass@1 is clearly higher than the Pass@2 of initial solutions. This shows that the model’s self-debugging performance is bad without training. The prompting approaches proposed by existing works such as (https://arxiv.org/pdf/2306.09896 and https://arxiv.org/abs/2304.05128) are not as effective on open-sourced LLMs. **This experiment further supports our motivation to train LLMs to self-debug and proves the effectiveness of our approach.**

| After SFT | | MBPP+ | HumanEval+ |
|-|-|-|-|
| StarCoder-15B | Pass@2 | 51.19 | 39.29 |
| | Expl. + Ref. Pass@1 | **53.83** | **43.54** |
| CodeLlama-7B | Pass@2 | 50.93 | 40.95 |
| | Expl. + Ref. Pass@1 | **51.55** | **47.62** |
| CodeLlama-13B | Pass@2 | 50.93 | 44.78 |
| | Expl. + Ref. Pass@1 | **54.59** | **51.32** |

---

### Decision · Program_Chairs · 2024-09-25

**Decision:**

Accept (poster)

**Comment:**

The paper proposes a framework to improve self-debugging for code generation tasks through a combination of SFT and RL using failed tests, their fixes coupled with explanations. The reviewers unanimously recommend its acceptance. They note strong empirical results obtained using comprehensive analysis done with various datasets and models, highlight its novelty for the reward setup and the importance of the problem. I therefore recommend its acceptance.